# Current status on the need for improved accessibility to climate models code

Juan A. Añel[1], Michael García-Rodríguez[1,2], and Javier Rodeiro[2]

[1]EPhysLab & CIM-UVigo, Ed. Campus da Auga, Campus As Lagoas, 32004, Ourense, Galicia (SPAIN)
[2]School of Computer Sciences, Campus As Lagoas, 32004, Ourense, Galicia (SPAIN)

**Correspondence:** Juan A. Añel (j.anhel@uvigo.es)

**Abstract.** Over the past few years, increasing attention has been focused on the need to publish computer code as an integral part of the research process. This has been reflected in improved policies on publication in scientific journals, including key related issues such as repositories and licensing. We explore the state-of-the-art of code availability and sharing of climate models, using as a testbed the models from the Fifth Coupled Model Intercomparison Project (CMIP5), and we include some particular reflections on this case. Our results show that there are many limitations in terms of access to the codes of these climate models and that the climate modelling community needs to improve its code-sharing practice to comply with best practice in this regard, and the most recent editorial publishing policies.

*Copyright statement.* TEXT

## 1 Introduction

Reproducibility of results is essential to comply with the scientific method when performing research. This has extraordinary implications in the field of earth system models (Añel, 2019; Gramelsberger et al., 2020). Because so much scientific output today relies on the use of computers, there are new requirements in terms of the description of any experiments performed, to assure computational scientific reproducibility (CSR). CSR, as defined by the U.S. National Academies of Science, Engineering, and Medicine, means 'obtaining consistent results using the same input data, computational methods, and conditions of analysis' (National Academies of Sciences, Engineering, and Medicine, 2019). This is widely known (Añel, 2011) and was recently discussed in a Sackler Colloquium on "Reproducibility of Research: Issues and Proposed Remedies" (Allison et al., 2018). CSR is a problem of high complexity.

In some cases, scientists may be unaware of some of the issues that have significant impacts on it, reaching the wrong conclusion that an experiment complies with CSR, when it is not the case, as exposed in Añel (2017). Also, a researcher could decide to use a model based on judgements that have little to do with the most appropriate from a scientific point of view, such as complying with the scientific method (Joppa et al., 2013). All this makes it necessary to consider a range of issues to comply with CSR in the process of design and use of models (Añel, 2017), and climate change models in particular. Some of

them are legal aspects of software distribution and intellectual property, usually unfamiliar for researchers. Recent examples have revealed some very low levels of CSR (Allison et al., 2018; Stodden et al., 2018). Steps are being taken to improve CSR, e.g., an increasing number of journals now have computer-code policies (Stodden et al., 2013; GMD Executive Editors, 2015; Nature, 2018), and recommendations have been made to ensure greater reproducibility of results (Wilson et al., 2017). They

include to maintain appropriate documentation for the software, split the code into functions and submit it to DOI-issuing repositories, encourage the participation of external collaborators and make it easy for them to collaborate, etc.

The study of climate change relies heavily on the use of large computer simulations with geoscientific models of varying levels of complexity. In projects involving the intercomparison of climate models and in some research papers, it has become increasingly common to provide details of the simulations performed. These details include initial configurations, which are

generally clear, accessible and formalised, in related outputs with digital object identifiers (DOIs) (e.g. Eyring et al. (2016); Morgenstern et al. (2017)). However, it is somewhat perplexing that the codes of the underlying models are not always made available. At best they are shared informally, using links, repositories without any security regarding long-term availability or access, or email addresses via which it is claimed that the code will be delivered after contact. Especially in a field where heated debates occasionally arise following the publication of results, it seems odd that this core element of the research is not

made more widely accessible.

There are other reasons that justify the need for access to the codes of climate models used in scientific research. One is to prevent the loss of knowledge on the cycles of development of these models. Some of them nowadays rely on 'legacy' code that was written up to five decades ago, and new developers must understand why some decisions on implementation were undertaken so long ago. There is both an educational and practical dimension to this issue. In some cases, different

models share sections of code, but its development remains fairly obscure (Knutti et al., 2013). It can be argued that adequate documentation of the code and the model is not necessary to prevent a potential loss of knowledge if the code used in the models includes appropriate comments. But, indeed, this is not the case of the models contributing to the Fifth Coupled Model Intercomparison Project (CMIP5). CMIP models are sophisticated software projects, and they need full documentation of the experiments (Pascoe et al., 2020).

Moreover, it is the case that climate models do not comply with what would be the ideal level of programming practice (i.e., coding standards, number of comments, documentation, etc.), an idea already pointed out by Wieters and Fritzsch (2018). García-Rodríguez et al. (2020) show how programmers have tended to perform very poorly in this regard in particular, and the incidence of comments throughout the code of some CMIP5 models is very low. Another issue related to the need for code sharing of climate models is the replicability of results. In different computing environments can also be challenging and

should not be expected by default (Easterbrook, 2014), even where the same model is used (Massonnet et al., 2019).

Some informal efforts have been made to document accessibility for some climate models (Easterbrook, 2009; RealClimate.org, 2009) and others more formally to check their quality (e.g., Pipitone and Easterbrook (2012); García-Rodríguez et al. (2020)). In light of these efforts, in this study, we intended to test the current status of accessibility to the most commonly used global climate models, in particular those that have contributed to the CMIP5. In the sections that follow, we describe

our efforts to gain access to these models, the procedures we followed, and a classification of the models according to some metrics related to accessibility, and we also provide a discussion containing reflections on the state-of-the-art.

## 2 Methods

In our attempt to better understand the current status of CSR and the availability of climate models, we used as a testbed the
models of CMIP5 (Taylor et al., 2012) given their extensive use in climate research over the last five years. These served as a vital tool for the last IPCC AR5 (IPCC, 2013), and given the ongoing development of CMIP6, groups of modellers should now be more open to sharing the code, due to the possible depreciation of the earlier version. We followed a standard procedure to obtain the code of each model. First, we checked the information available on each model in the webpage of CMIP5. Then we contacted research groups where necessary using email, without disclosure of ourselves as climate scientists to full explanation
of our interest in studying the code. Our approach is detailed in the following sections.

### 2.1 Survey methods

Using a systematic methodology, we attempted to obtain the codes of all the climate models involved (see Table 1). This procedure included the first step using the web addresses for code downloading indicated on the CMIP5 webpage (https://pcmdi.llnl.gov/?cmip5/). When the code was not directly available to be downloaded, we looked for contact details (emails,
on-line formularies, etc.) in the webpage. In some cases, there was an email address to contact. However, in other cases following the information in the CMIP5 website was not enough. In such cases, we searched through the internet using a search engine. We looked for institutional web pages, intending to find open repositories for the corresponding model. In a few cases, this was sufficient (see Table 1). Still, in others, we had to proceed by making contact with development teams at different levels, e.g. emails (see Appendix) (with follow-up emails two weeks after the first contact). In the case of the IPSL
team after sending the email in English and failing to get a reply, we sent the second email in French, and we got an answer. For the NASA-GMAO model, we were unable to get a response from contact via email. However, we were approached by a member of the institution after a presentation during a conference. After discussing it, the development team granted us access to the model.

For those cases where we needed to establish contact via email, we provide details in the Table 1 of the different replies
that we received. Michael García-Rodríguez always sent the first email (see Appendix A1) from his student address (under the domain esei.uvigo.es), who had had no previous involvement in the activities of the international climate modelling community. The idea behind this was to check whether after it had become evident that the models were not available easily, institutions and researchers would then share them with someone from outside the community. In the end, to assure CSR and accessibility, details of experiments must be open to everybody, not just to peers or other scientists. A second email, equal to the first one
(Appendix A1), was sent to insist on our request. We intended to minimize the possibility of not getting a reply because of reasons such as the contact person being too busy at the moment of receiving the first email, it was unnoticed or filtered as spam. We waited for a reply for three weeks after sending the first email before sending the second. Finally, three weeks after

the second email, we sent a final email (Appendix A2), where we identified ourselves and our team, to make clear that we were indeed climate scientists, and thus to check whether we thus had a better chance of obtaining the code. Where access to the code was denied, we sent a survey with a few questions to better understand the reasons for this. All emails sent followed the same template as that given in the Appendix A.

It could be possible to look for additional contact information in the published scientific literature. However, many papers continue to be accessible only through paywalls (they are not open-access) and therefore they are not available for most of the people. Moreover, identifying the relevant person to contact from an author list requires a knowledge of the modelling groups that only a handful of experts in the field have. Also, additional contact information is available for another five models from metadata included in the NetCDF files containing the results of the simulations in the CMIP5 repository. However, the ability to

find and manage such data has computational requirements and needs of a knowledge that is beyond what could be considered reasonable for the general public, including part of the scientific community.

## 3    Results

After all attempts and several months, we successfully gained access to 10 out of 26 models (27 out of the 61 model versions or configurations) contributing to CMIP5. Table 2 provides a summary of the details of the replies obtained from these centres,

teams or contacts that allowed access to the code. In terms of research centres or groups contributing to the CMIP5 project, this also represents 10 out of a possible 26. We found a strong regional bias in terms of the countries where models were made accessible. The USA, Germany and Norway stood out as the best contributors in that we obtained the code for all their models (though Norway only contributes one). Together, these three countries represent 38% of the research centres or CMIP5 models and 44% of all the versions. For France, we gained access to one of two models (three out of five versions). We can speculate

that in some cases, the decision on whether to share the code of the models, could have been influenced by national or regional regulations on software copyright, intellectual property, etc. For example, it is well known that the law in the USA, where for instance software can be patented, enables the possibility to enforce a higher level of restrictions to software sharing and distribution than in the EU (van Wendel de Joode et al., 2003; European Patent Office, 2016). However, it is the case that we have been able to get the code for 6 of the 7 models contributed by research centres from the USA and yet, for the EU we have

only got 3 in 7. The fact that the models developed in the USA can count with the participation of federal employees could partially explain this result. Under the USA copyright law (U.S. Code, 1976), all the work produced by federal employees is in the public domain. Although they are not the same thing, the public domain could be considered closer to openness that lack of licensing of the models. For the case of Norway, we can speculate that the fact that NorESM has been developed using core parts of CESM1 (Knutti et al., 2013) could have facilitated openness of the code through the inheritance of licenses and

copyright. In the same way, not sharing the code of the models could be due to inheritance reasons.

     Figure 1 shows the percentage of models obtained from a global perspective, with specific plots for Europe and Asia. This makes it easier to visualise the rather narrow distribution of the regions on the maps and because different countries could apply different national laws in order to share the codes of the models.

In some cases, a high number of email exchanges were required over periods longer than one week to receive a reply or the code. In five cases, there was no obvious way to contact the development teams; in four cases, we received no answer at all. Seven research centres (corresponding to eighteen models) they replied that they did not share the codes of their models. We decided to include in this final group EC-Earth, for which the code is said to be available to a given group of users. Still, in practice, the procedure to access it makes it completely unfeasible for non-members of the regular team involved in its development. In no case did we receive a response to the questionnaire sent asking for the reasons why they did not want to share the code. For the models obtained, we performed a ranking, as shown in Table 3, taking into account licensing issues and availability for reuse by third parties, among other factors. We considered the level of requirements introduced by the GPLv3 license (https://www.gnu.org/licenses/gpl-3.0.en.html) as the ideal case for a license under which the model can be shared, modified and used without restriction. This is in line with the recent updates to the policy on code availability published by Geoscientific Model Development (GMD Executive Editors, 2019). Moreover, it has been argued that it is the license that better fits to scientific projects to assure the benefits and openness of software (Morin et al., 2012).

We also addressed other issues relevant for running the models. In some ways, accessibility or ability to gain access to the code means nothing if adequate documentation for the model, a description of its components, instructions on how to compile or run it, and basic examples are not provided. This is in line with recommendations contained in the literature (Lee, 2018). The results are shown in Table 4. It can be seen that almost all the models obtained comply with all these criteria, except for NICAM.09, which only includes a 'Readme' and a 'Makefile'. For the IPSL, although the link to access the documentation does not work, it is possible to gain access to it by performing an internet search.

## 4 Conclusions

In this work, for cases where we obtained the code of a given model, we were not provided with a reason for the license behind it. In fact, in some cases, despite getting the code, we did not see a license explaining clearly the terms of use. Some scientists or model development groups could be worried because of issues such as legal restrictions (national or institutional) that prevent from publishing code. Also, because of potential dependencies of the model on third party proprietary software, lack of funding to maintain a public repository or violation of property rights. For all these cases, there is a clear response or solution:

- In the first case, if it is not possible to make available the code, then any result obtained with such a model should not be accepted as scientifically valid because it is impossible to verify the findings. We acknowledge that this is the case for several models widely used in scientific research nowadays, and this situation must be solved by modification of the legal precepts applied to them. Consequently, those working with such models should look for a change in the legal terms so that the model complies with the scientific method.

- If a part of the model depends on proprietary software, then it harms the possibility to distribute the whole model. Therefore again, the model does not comply with the scientific method. In this case, a good option can be to substitute the part of the model that is proprietary software with one that is free software.

– Lack of funding to maintain a software repository can not be considered a real problem. There are many options available to host the code. For example, Zenodo is free, widely adopted and assures hosting at least for the next twenty years.

– Fears about a violation of property rights usually respond more to a lack of awareness on how the law applies to software distribution than to real issues, as Añel (2017) points out. Intellectual property is usually detached of the norms that apply to the distribution of the model. Unless the developer specifically resigns to the intellectual property, it is generally retained despite if the software is made available and distributed and their contractual obligations. Indeed, under some legal frameworks, it is impossible to resign to intellectual property. The best option is always to get specialised legal advice on these matters.

It is a matter of some regret that we obtained straightforward access to just 3 of the 26 models (7 of the 61 versions) in CMIP5 and that for 16 (34 versions) we were not able to obtain the code at all. For all others, some interaction was required, from email exchanges to personal discussions at workshops. Indeed, we did not get access to the codes for more than half of all the versions used in the CMIP5 despite identifying ourselves as research peers. Therefore, we have to report the very poor status of accessibility to climate models, which could generate serious doubts for the reproducibility of the scientific results produced by them. While there is no reason to doubt the validity of the results of the study of climate change obtained using the CMIP5 models (in a similar way to findings for other disciplines (Fanelli, 2018)), we encourage all model developers to improve the availability of the codes of climate models and their CSR practices. Previous work has already shown that there is room for significant improvement in the structure of the codes of the models, which is in some cases very poor (García-Rodríguez et al., 2020), and sharing it could help to alleviate this situation. It would be desirable that future efforts on the development of climate models have into account the results here presented. In this vein, scientists starting model development from scratch, without problems with legacy code, are in a great position to care from the beginning about licensing and reproducibility, doing it in the best way possible.

It is possible to speculate that some scientists could be reluctant to share code because of perceived potential damage to their reputations because of its quality (Barnes, 2010). It can be argued that this could be related to a lack of adequate education in computer programming (Merali, 2010). This issue with the quality of the code has been raised for the case of climate models (García-Rodríguez et al., 2020). Given that many scientists have no formal training as programmers, it may be presumed that they consider that their code may not comply with the standards of excellence that they usually pursue in their primary fields of knowledge. Indeed, it has been documented that some climate scientists acknowledge that imperfections in climate models exist, and they address them through continuous improvement without paying too much attention to the common techniques of software development (Easterbrook and Johns, 2009).

Nevertheless, all scientists must believe that their code is good enough (Barnes, 2010) and that there are thus no reasons not to publish it (LeVeque, 2013). Barriers to code-sharing exist through licensing, imposed by, e.g., government bodies, or propriety considerations. They are not due to technical difficulties or scientific reasons. When contributing to scientific studies and international efforts where collaboration and trust are critical, such practice limits the reproducibility of the results (Añel, 2019).

We recommend that frozen versions of the climate models later used to support the results discussed in international reports on climate change should be made accessible along with the outputs from simulations in official data portals. Also, this should apply to any other Model Intercomparison Project. It must be considered that climate models are an essential piece in the evaluation of climate change, and not sharing them can be perceived as a weakness of the methodology used to perform such assessments. Reproducibility could not be compromised; however, the simple fact that replicability (note that reproducibility and replicability are different concepts (ACM, 2018)) can not be achieved because of the lack of the code of the model is unfortunate. In this way, frozen versions of the models combined with cloud computing solutions and technologies such as containers can be a step forward to achieve full replicability of results in earth system modelling (Perkel, 2019; Añel et al., 2020). Also, tools to validate climate models are becoming common. Such tools use metrics to validate the outputs of the models. The ESMValTool (Righi et al., 2020) has been designed with this purpose and evaluation of the accessibility and code of the model could be integrated as a part of the process to measure the performance of the models contributing to the CMIP.

An additional reason to request an open code software policy is that several scientific gaps have been pointed out for the CMIP5 (Stouffer et al., 2017). The lack of availability of the code of the models makes it difficult to address them, as it is not possible to perform a complete evaluation of the source of discrepancies between them. As it has been shown for other fields of software development, sharing the code can help to improve the development process of climate models and their reliability (Boulanger, 2005; DoD CIO, 2009). Moreover, it would help to support the collaborative effort necessary to tackle the challenge of climate change (Easterbrook, 2010) and to do it in a way that complies better with the scientific method and the goals of scientific research (Añel, 2019). Funding should be allocated by agencies and relevant bodies to support such efforts. Notwithstanding that the whole framework of science faces challenges related to CSR, at the same time presents opportunities for improvement in such a sensitive field as climate science.

*Code and data availability.*  There is no code or data relevant to this paper.

**Figure 1.** Geographical map with the number of the models obtained for each country: a) worldwide; b) Europe (EC-Earth is only included in the worldwide view because it is developed as a consortium of sixteen European countries); c) Asia. Green colours and fractions represent the obtained models from the total.

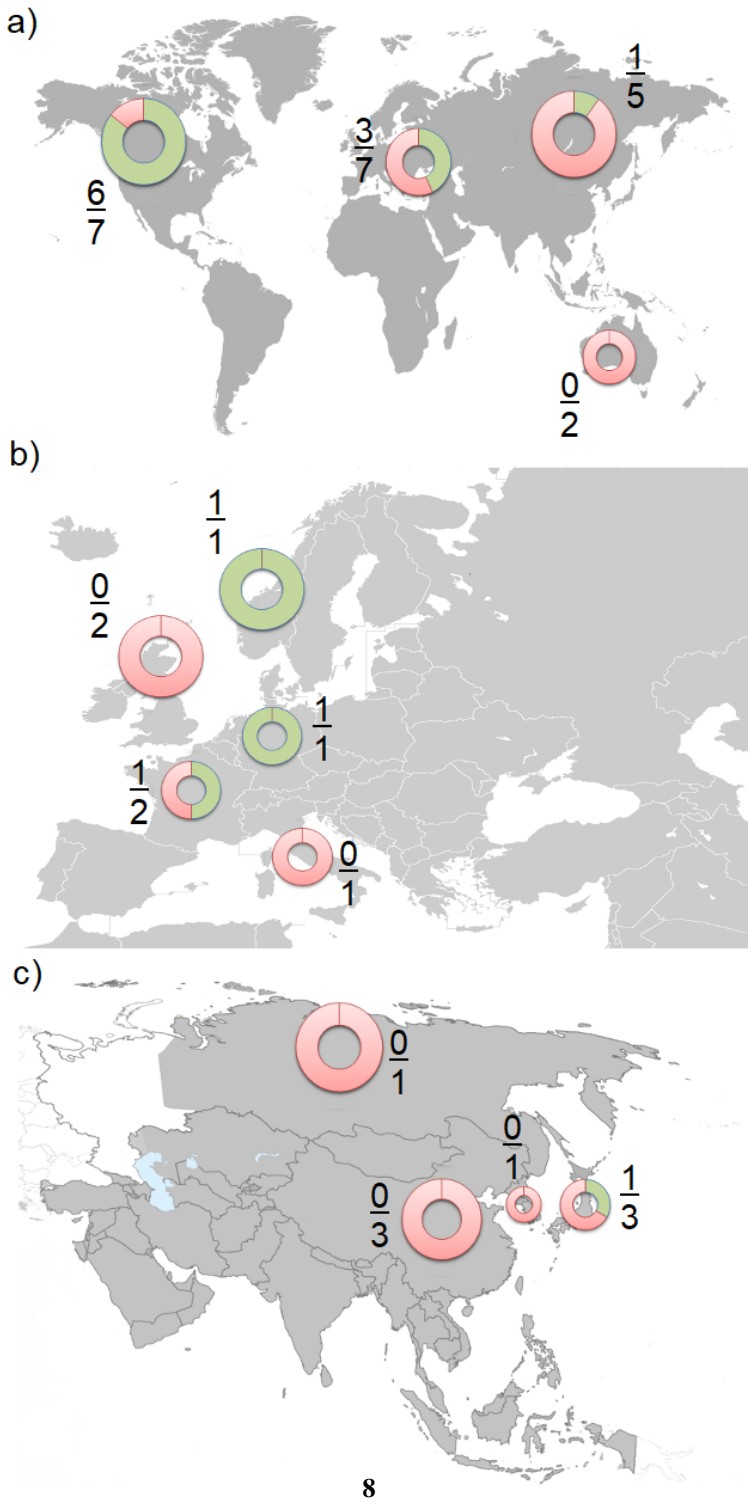

Table 1: CMIP5 model list, research centre responsible for each one and details on the procedure for accessing their code. Emails 1 and 2 can be seen in the Appendix A1. Email 3 (Appendix A2) is not listed because we did not receive any answer to them.

| Modeling center | Model | Free download | Answer Email 1 | Answer Email 2 | Comments/Answer |
|---|---|---|---|---|---|
| BCC | BCC-CSM1.1 | No | - | - | No email or contact phone is available. |
| | BCC-CSM1.1(m) | No | | | |
| CCCma | CanAM4 | No | Yes | | The code is not shared. |
| | CanCM4 | No | | | |
| | CanESM2 | No | | | |
| CMCC | CMCC-CESM | No | No | No | No answer. |
| | CMCC-CM | No | | | |
| | CMCC-CMS | No | | | |
| CNRM-CERFACS | CNRM-CM5 | No | No | Yes | The code is not shared. |
| | CNRM-CM5-2 | No | | | |
| COLA and NCEP | CFSv2-2011 | Yes | - | - | Code available from the official web site. |
| CSIRO-BOM | ACCESS1.0 | No | Yes | | The code is not shared. |
| | ACCESS1.3 | No | | | |
| CSIRO-QCCCE | CSIRO-Mk3.6.0 | No | - | - | No email or contact phone is available. |
| EC-EARTH | EC-EARTH | No | - | - | The code is not shared. |
| FIO | FIO-ESM | No | No | No | No answer. |
| GCESS | BNU-ESM | No | No | No | No answer. |
| INM | INM-CM4 | No | - | - | No email or contact phone is available. |
| IPSL | IPSL-CM5A-LR | Yes | Yes | | Available after email exchange. |
| | IPSL-CM5A-MR | Yes | | | |
| | IPSL-CM5B-LR | Yes | | | |
| LASG-CESS | FGOALS-g2 | No | No | No | No answer. |
| LASG-IAP | FGOALS-gl | No | - | - | No email or contact phone is available. |
| | FGOALS-s2 | No | | | |
| MIROC | MIROC4h | No | Yes | | The code is not shared. |
| | MIROC5 | No | | | |
| | MIROC-ESM | No | | | |
| | MIROC-ESM-CHEM | No | | | |

| Institute | Model | | | | Comment |
|---|---|---|---|---|---|
| MOHC | HadCM3 | No | No | Yes | The code is not shared. |
| | HadCM3Q | No | | | |
| | HadGEM2-A | No | | | |
| | HadGEM2-CC | No | | | |
| | HadGEM2-ES | No | | | |
| MPI-M | MPI-ESM-LR | Yes | Yes | - | Available after email exchange. |
| | MPI-ESM-MR | Yes | | | |
| | MPI-ESM-P | Yes | | | |
| MRI | MRI-AGCM3.2H | No | - | - | No email or contact phone is available. |
| | MRI-AGCM3.2S | No | | | |
| | MRI-ESM1 | No | | | |
| | MRI-CGCM3 | No | | | |
| NASA-GISS | GISS-E2-H | Yes | No | No | Available after email exchange. |
| | GISS-E2-H-CC | Yes | | | |
| | GISS-E2-R | Yes | | | |
| | GISS-E2-R-CC | Yes | | | |
| NASA-GMAO | GEOS-5 | Yes | No | No | Available after meeting during a workshop. |
| NCAR | CCSM4 | Yes | - | - | Code available from the official web site. |
| NCC | NorESM1-M | Yes | Yes | | Available after email exchange. |
| | NorESM1-ME | Yes | | | |
| NICAM | NICAM.09 | No | No | Yes | Available after email exchange. |
| NIMR/KMA | HadGEM2-AO | No | No | Yes | The code is not shared. |
| NOAA-GFDL | GFDL-CM2.1 | Yes | Yes | Yes | Available after email exchange. |
| | GFDL-CM3 | Yes | | | |
| | GFDL-ESM2G | Yes | | | |
| | GFDL-ESM2M | Yes | | | |
| | GFDL-HIRAM-C180 | Yes | | | |
| | GFDL-HIRAM-C360 | Yes | | | |
| NSF-DOE-NCAR | CESM1(BGC) | Yes | - | - | Code available from the official web site. |
| | CESM1(CAM5) | Yes | | | |
| | CESM1(CAM5.1,FV2) | Yes | | | |
| | CESM1(FASTCHEM) | Yes | | | |
| | CESM1(WACCM) | Yes | | | |

Table 2: Summary of reasons behind granting us access to the source code of the models.

| Modeling center | Model | Process and reasons to access to the code |
|---|---|---|
| COLA and NCEP | CFSv2-2011 | A tarball with the source code can be easily accessed from the official web site explaining what the code does and how the climate model works. |
| IPSL | IPSL-CM5A-LR | M. García-Rodríguez identified himself and explained via email the purposes of this research. After a meeting of the developing team and additonal details on this research we were granted access to a tarball with the source code. |
| | IPSL-CM5A-MR | |
| | IPSL-CM5B-LR | |
| MPI-M | MPI-ESM-LR | The access to a tarball with the source code was granted after registration as an user via a web page and approval, without any extra communication or reasoning. |
| | MPI-ESM-MR | |
| | MPI-ESM-P | |
| NASA-GISS | GISS-E2-H | After two weeks, we received the answer to our email. They provided us with a link to a tarball with the source with the snapshots of the model. |
| | GISS-E2-H-CC | |
| | GISS-E2-R | |
| | GISS-E2-R-CC | |
| NASA-GMAO | GEOS-5 | Initially, they did not answer the emails that were sent to them. After a presentation during a workshop, Dr Añel was approached by one of the team members. He put us in contact with one of the coders. We obtained access by contacting this person. The code was available as 4073 files in directories retrieved using 'wget'. |
| NCAR | CCSM4 | The code of the model is available through a web page. The download proccess is open to anyone but it is hard. Each file of the model has to be individually retrieved (2247 files in total, each in its respective sub-directory). |
| NCC | NorESM1-M | First, we received a reply stating that the code of the model is not shared with anyone outside the NorESM-community, asking if we really needed it. After identifying ourselves and explaining our research, we were granted access to a tarball after registering as users in the 'noresm wiki'. |
| | NorESM1-ME | |
| NICAM | NICAM.09 | Initially, they asked us questions about the purpose of obtaining the code. Then, after explaining the objectives of the project, they granted us access to a tarball with the code. We had to register in the NICAM user group. |
| NOAA-GFDL | GFDL-CM2.1 | We were granted access to a tarball with the source code in reply to our first request via email. |
| | GFDL-CM3 | |
| | GFDL-ESM2G | |
| | GFDL-ESM2M | |
| | GFDL-HIRAM-C180 | |

| | GFDL-HIRAM-C360 | |
|---|---|---|
| NSF-DOE-NCAR | CESM1(BGC) | We had to register to access the Community Earth System Model. After that, we were able to download a tarball with the source code. |
| | CESM1(CAM5) | |
| | CESM1(CAM5.1,FVV2) | |
| | CESM1(FASTCHEM) | |
| | CESM1(WACCM) | |

Table 3: CMIP5 models with code obtained and scores of reproducibility. Maximum value of three filled stars is given to those models it is possible to access through the internet without restriction, with a license that allows full testing and evaluation of the model. The score was reduced by one star when failing for each one of the following criteria: if in order to gain access to the model we had to contact a research centre or development group, to sign license agreements, or if we gained access only after identifying ourselves as scientists undertaking climate research and according to the rights to evaluate and use the model as granted by the license (if applicable). A not-filled star means that the license of the model does not allow modification of the code.

| Institution | Model | Score |
|---|---|---|
| Cola & NCEP | CFSv-2011 | ★★☆ |
| IPSL | IPSL-CM5A-LR<br>IPSL-CM5A-MR<br>IPSL-CM5B-LR | ★★ |
| MPI-M | MPI-ESM-LR<br>MPI-ESM-MR<br>MPI-ESM-P | ★ |
| NASA GISS | GISS-E2-H<br>GISS-E2-H-CC<br>GISS-E2-R<br>GISS-E2-R-CC | ★☆ |
| NASA GMAO | GEOS-5 | ★☆ |
| NCAR | CCSM4 | ★★★ |
| NCC | NorESM1-M<br>NorESM1-ME | ★★ |
| NICAM | NICAM.09 | ★ |
| NOAA GFDL | GFDL-CM2.1<br>GFDL-CM3<br>GFDL-ESM2G<br>GFDL-ESM2M<br>GFDL-HIRAM-C180<br>GFDL-HIRAM-C360 | ★★ |
| NSF-DOE-NCAR | CESM1(BGC)<br>CESM1(CAM5)<br>CESM1(CAM5.1 FV2)<br>CESM1(FASTCHEM)<br>CESM1(WACCM) | ★★★ |

Table 4: Availability of detailed information provided with the source code of the models in order to run them. 'Documentation' refers to full documentation of the model (for IPSL models a web address/link was included to access the documentation but it did not work). 'Readme' corresponds to a file containing basic explanations on the files part of the model and basic instructions. 'Basic example' refers to whether an example to explain the model is included. 'Dependencies' refers to the basic information on libraries, compilers or any other software and its version needed to run the model. 'Makefile' refers to the existence of a single file that manages all the process of compilation and model run.

| Modeling center | Model | Documentation | ReadMe | Basic example | Depencencies listed | Makefile |
|---|---|---|---|---|---|---|
| COLA and NCEP | CFSv2-2011 | yes | yes | yes | yes | yes |
| IPSL | IPSL-CM5A-LR | no* | yes | yes | yes | yes |
| | IPSL-CM5A-MR | | | | | |
| | IPSL-CM5B-LR | | | | | |
| MPI-M | MPI-ESM-LR | yes | yes | yes | yes | yes |
| | MPI-ESM-MR | | | | | |
| | MPI-ESM-P | | | | | |
| NASA-GISS | GISS-E2-H | yes | yes | yes | yes | yes |
| | GISS-E2-H-CC | | | | | |
| | GISS-E2-R | | | | | |
| | GISS-E2-R-CC | | | | | |
| NASA-GMAO | GEOS-5 | yes | yes | yes | no | yes |
| NCAR | CCSM4 | yes | yes | no | yes | yes |
| NCC | NorESM1-M | yes | yes | yes | yes | yes |
| | NorESM1-ME | | | | | |
| NICAM | NICAM.09 | no | yes | no | no | yes |
| NOAA-GFDL | GFDL-CM2.1 | yes | yes | yes | yes | yes |
| | GFDL-CM3 | | | | | |
| | GFDL-ESM2G | | | | | |
| | GFDL-ESM2M | | | | | |
| | GFDL-HIRAM-C180 | | | | | |
| | GFDL-HIRAM-C360 | | | | | |
| NSF-DOE-NCAR | CESM1(BGC) | yes | yes | yes | yes | yes |
| | CESM1(CAM5) | | | | | |
| | CESM1(CAM5.1,FVV2) | | | | | |
| | CESM1(FASTCHEM) | | | | | |

| | CESM1(WACCM) | | | | | |
|---|---|---|---|---|---|---|

### Appendix A:  Templates of emails used to contact the model development teams

### A1    First email

Dear Sir/Madame,

my name is Michael García Rodríguez and I am an MSc Student at the EPhysLab in the Universidade de Vigo, Spain

(http://ephyslab.uvigo.es). I am developing my MSc Thesis on the study of qualitative issues of climate models, mostly re-

lated to scientific reproducibility and copyright issues.

In order to do it, I have focused my research project on the study of the models that contributed to the last CMIP5 report. For

it, I am trying to get access to the code of all the models that reported results of this effort.

Therefore I would like kindly request access to the code of your model, $MODEL - NAME$, namely the version that you

used to produce CMIP5 results. Therefore, could you say me how could I get access to it?

Many thanks in advance.

Best regards,

Michael García Rodríguez

EPhysLab

Universidade de Vigo

http://ephyslab.uvigo.es

========================

### A2    Third email

Dear Sir/Madame,

Two weeks ago I send you the following email:

(see email in Appendix A1)

Would you kindly answer me ? In case I am not allowed to access the code, could you explain me why? It would be of great

help, in case of not being able to get the code of the model, know the answer. Please, if it's possible, mark with a cross one or

more answers on below:

[ ] Copyright issues (please, if you mark this choice, could you send me a copy of the licenses?)

[ ] Development team policy

[ ] Legal restrictions of your country

[ ] Others reasons (please specify):

_________________________________________-

In this case, I will be able to write down the reasons why I was not allowed access to the code and I could document it in my MSc Thesis on the study of qualitative issues of climate models.

Many thanks in advance,

5 Best regards,

Michael García Rodríguez

EPhysLab

Universidade de Vigo

10 http://ephyslab.uvigo.es

========================

*Author contributions.* All the authors participated in the design of the study and writting of the text. MGR and JAA made the attemps to get access to the code of climate models.

*Competing interests.* We do not have competing interest.

15 *Acknowledgements.* This research was partially supported by the ZEXMOD Project of the Goverment of Spain (CGL2015-71575-P) and the European Regional Development Fund (ERDF). Juan A. Añel was supported by a 'Ramón y Cajal' Grant funded by the Government of Spain (RYC-2013-14560). We would like to thank to Didier Roche, Julia Hargreaves, Rolf Sander, Richard Neale and two anonymous referees for useful comments to improve this paper.

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
