# Peer review of "Current status on the need for improved accessibility to climate models code"

_Geoscientific Model Development, 2019_

## Short Comment (SC1) · 8 Sep 2019

I just wanted to comment on the aim and transparency of the work in the paper.

The topic of this 'study' certainly has merit as modeling centers are trying to improve their transparency and availability of model code, but it is an ongoing process in the face of certain restrictions that were noted. I have significant concerns though, regarding its approach and methods and inappropriate speculation. One thing that stood out for me as complete speculation is the line "It is widely acknowledged that some scientists are reluctant to share code because of the perceived potential damage to their reputations". This is quite inappropriate for an academic journal, and I really do not understand what is meant by this. Being pretty intensively involved in CMIP efforts for

many years, I have never come across code release reluctance related to 'reputation'

Furthermore, we simply have no idea how rigorous the investigation is. It quoted a whole raft of comments from various staff at institutions that cannot be attributable to anyone. We have no idea whether they spoke to junio scientist or a senior director. And it is worth pointing out that a little deception was used in the attribution to a researcher as a PhD student in their emails.

Don't get me wrong, easily accessible access to code is important in the whole climate change and climate modeling sphere, but the authors need to rethink this approach quite a bit.

Of course I will let the handling editor and reviewers determine whether they agree with me :)

Thanks Rich Neale

---

## Referee Comment (RC1) · Anonymous Referee #1 · 10 Sep 2019

Review of "Current status on the need for improved accessibility to climate change models" by Juan A. Añel, Michael García-Rodríguez, and Javier Rodeiro.

The manuscript looks into the availability of the source code of models which contributed to CMIP5. The authors basically use a three-step approach to get access to the code (direct download, "anonymous" email and email stating their research). Their results show that more than half of the model source code is not made made available after step three. In addition, they discuss the documentation quality and licensing issues.

I find the topic of the manuscript to be highly relevant and the results presented by the authors raise a crucial issue which is of high importance for the climate model community. However, I find the manuscript to have several weaknesses which should

be addressed before a possible publication in Geoscientific Model Development as outlined in the comments below.

General comments:

**1 The manuscript contains several generalized statements which I find to need further support by (scientific) literature or the results. I've addressed them also in the specific comments but in general I refer to statements like the following: "...it is generally the case that climate models do not comply with what would be the ideal level of programming practice." (page 1, line 11) "...the incidence of comments throughout the code is very low" (p1, l13) "It is widely acknowledged that some scientists are reluctant to share code because of the perceived potential damage to their reputations." (p4, l14)**

**2 I find the discussion about the need to publish model code in the introduction too one-sided. While the authors give several good reasons why code should be made publicly available, there might be equally valid counter-arguments. It would be good to discuss some of them as well and possibly offer solutions. Some things that come to mind:**

- National or institutional copyright that prevents authors from publishing code

- Dependencies of the model on third party code that is under copyright

- A lack of funding to set up and maintain a public code repository

- Fear that ones property rights might be violated if the code is freely available in the web. This might be particularly true for newer models. An argument could be made that a group developing a model has the right to also publish the results produced with this model (they might even be required to do so by their funding agency).

**3 The way the authors tried to establish contact is not clearly enough documented and needs clarification. As it is, it seems quite subjective to me. In particular:**

- How did the authors search for contact information? They mention that they searched

the internet. But how easy was it to find the information and at what point (after how much time) did the authors give up? One could for example go as far as looking for publications by the same group investigating the desired model and writing the authors. I was quickly able to find contact information for all five model centres the authors list as "No email or contact phone is available" (see specific comments below).

- The two mails in A1 and A2 are identical. Is that a mistake or did the authors just send the same mail again? In this case they could consider deleting A2. In section 2.1 the authors mention that in "the final email" they identified themselves. After how many mails of type one was that? The "final email" should also be in the appendix. In table 1 only email 1 & 2 are listed. Are these two columns indeed refering to the identical emails from A1 & A2? If yes I'm missing the column for the final email.

- The authors mention one specific model for which they got access after direct contact to the developers at a conference. Was this approach tried for all model centres which did not reply to the emails or was it based on a coincidence? If it was only done in this one case, one could argue that this partly jeopardizes the objectivity of the approach.

**4 The manuscript would benefit from proof-reading by a native speaker. There are a number of very long and somewhat convoluted sentences. This sometimes makes it hard to follow the authors point as I mention at several occasions in my specific comments.**

Specific comments:

Title: Maybe change to "model code" in order to make clear that this is about the source code and not about output? Also I believe that it would be more appropriate to call them "climate models" instead of "climate change models" because they are used to investigate the climate system in general not only climate change.

Abstract: "models from the Climate Model Intercomparison Project" Add "fifth"

Page 1, line 13-16: This is a very long sentence and I'm not quite sure what the authors

try to say here. Is this addressing the issue of reproducability in general? E.g., subjective judgments (as long as they are properly documented) to not hinder reproducability. So maybe the authors try to say something else? Please clarify, since this seems to be an important point.

P1, l16: replace "matters" by "CSR"

P1, l19: It would be convenient for the reader to have the (most important) recommendations listed here, instead of only quoting Wilson et al. 2017.

P2, l11: "It could be said that adequate sharing and documentation is not necessary if the code used in the models includes appropriate comments," I'm not sure what the authors try to say here. Why would commenting the code make publishing unnecessary? Also I find the statement that commenting code can replace a proper documentation problematic.

P2, l12-15: "...but it is generally the case that climate models do not comply with what would be the ideal level of programming practice." I'd like to see some evidence to support this statement.

P2, l13-15: "Indeed, the incidence of comments throughout the code is very low, and programmers have tended to perform very badly in this regard in particular (García-Rodríguez et al., 2019)" What code to the authors refer to here? What does low number of comments mean? The citation (García-Rodríguez et al., 2019) is listed as submitted, it should be provided.

P2, l21: CMIP 5 I assume?

P2, l30 I wouldn't call two different emails a "variety of different approaches".

P3, l3: add "as a first step..."

p3, l3: Could the authors provide a link to the CMIP5 webpage they refer to here?

P3, l5: Why English and French? Several models are developed, e.g., in China it

seems to me that for a systematic and objective methodology it should be either only English (the language used in the vast majority of climate model related publications) or all respective native languages of the model centres.

P3, l11-12: "to check whether after it had become obvious that the models were not available easily, institutions and researchers would then share them with someone from the general public." To really represent the general public it would have been better to use a not-university related mail address I assume. Maybe change to something like "with someone from outside the community"?

P3, l18: Why did it take several months? The authors state above that they send an initial email with a follow-up two weeks later. Was it the process of actually getting the code after establishing contact that took so long?

P3, l18: To I assume correctly that "10 out of 26 models" referrers to "models from 10 out of 26 institutions" and "27 out of 61" referrers to multiple models from the same institution (such as GCM and ESM versions)?

P3, l23: The percentages refer to the numbers in the first line of the paragraph, I assume. The "they" seems to indicate that only USA, Germany, and Norway are meant.

P3, l24-26: "This analysis is relevant, because in some cases the decision on whether to share the code of the models could have been due to national or regional regulations on software copyright, intellectual property, etc." It would be interesting to detail this point further. Did the authors get concrete answers citing (national) copyright law as response? Did they check the copyright law in countries where they did not get access to any models?

P3, l30: "in six cases" & "in five cases" Table 1 only lists 5 cases with "No email or contact phone is available." and only 4 cases with "No answer".

P4, l3-5: "We considered the level of requirements introduced by the GPLv3 license (https://www.gnu.org/licenses/gpl-3.0.en.html) as the ideal case, or a license under

which the model can be shared, modified and used without restriction." This seems to be contradictory. Is the GPL the ideal case or "a license under which the model can be shared, modified and used without restriction"? Because to my knowledge the GPL has very strict requirements that need to be fulfilled to share, modify and use code which is licensed under it (such as: disclosure of source, stating of all changes, further publication only under the same license).

P4, l14-15: "It is widely acknowledged that some scientists are reluctant to share code because of the perceived potential damage to their reputations." Can the authors provide some evidence for this statement?

P4, l20-23: "Barriers to code-sharing through licensing, imposed by e.g., government bodies, cannot be an excuse and when contributing to scientific studies and international efforts where collaboration and trust are critical, such practice is not acceptable." It seems to me that following the laws of ones country (even when they hinder research and collaboration) is indeed a completely valid excuse for not sharing code.

P4, l22-24: "For cases where we obtained the code of a given model, we were not provided with a reason for the license behind it. In fact, in some cases despite getting the code we did not see a license explaining clearly the terms of use." It would be interesting indeed to know the rationale behind different licenses (or for the absence of a license), did the authors inquire about this at the groups which provided code?

P4, l31-32: "we encourage all model developers to improve the availability of the codes of climate models and their CSR practices." As I've mentioned before this might not be only the developers responsibility but also includes institutions, funding agencies and even country copyright laws.

P4, l33: "which is in some cases very poor (García-Rodríguez et al., 2019)" Again, please provide this paper as it is not yet available.

Figure 1a: Maybe delete the axis? (or fix the y-axis, which should run from -90 to

[Figure]

90 I assume?) The percentages are not per country as stated by the caption but by continent I assume? Could the authors find a way to explicitly show that there are now model centres in South America and Africa?

Figure 1: I personally would find it more helpful to see absolute numbers of provided/not provided models instead of percentages.

Table 1: I checked our CMIP5 archive and found all of the contact information flagged as missing in the metadata of the output netCDF files for the respective models. I understand that at this point it is probably too late to include them in the study, yet I'm copying them in here in case they are helpful to the authors.

BCC: "contact = "Dr. Tongwen Wu (twwu@cma.gov.cn)""

CSIRO-Mk3.6.0: "contact = "Project leaders: Stephen Jeffrey (Stephen.Jeffrey@qld.gov.au) & Leon Rotstayn (Leon.Rotstayn@csiro.au). Project team: Mark Collier (Mark.Collier@csiro.au: diagnostics & post-processing), Stacey Dravitzki (Stacey.Dravitzki@csiro.au: post-processing), Carlo Hamalainen (Carlo.Hamalainen@qld.gov.au: post-processing), Steve Jeffrey (Stephen.Jeffrey@qld.gov.au: modeling & post-processing), Chris Moeseneder (Chris.Moeseneder@csiro.au: post-processing), Leon Rot-stayn (Leon.Rotstayn@csiro.au: modeling & atmos. physics), Jozef Syk-tus (Jozef.Syktus@qld.gov.au: model evaluation), Kenneth Wong (Ken-neth.Wong@qld.gov.au: data management), Contributors: Martin Dix (Mar-tin.Dix@csiro.au: tech. support), Hal Gordon (Hal.Gordon@csiro.au: atmos. dynamics), Eva Kowalczyk (Eva.Kowalczyk@csiro.au: land-surface), Siobhan O\'Farrell (Siobhan.OFarrell@csiro.au: ocean & sea-ice)""

INM-CM4: "contact = "Evgeny Volodin, volodin@inm.ras.ru,INM RAS, Gubkina 8, Moscow, 119333 Russia,+7-495-9383904""

LASG-IAP: ":contact = "Dr. Tianjun Zhou(zhoutj@lasg.iap.ac.cn)" "

MRI: "contact = "Seiji Yukimoto (yukimoto@mri-jma.go.jp)""

Table 3: I find the star-rating system slightly in-transparent. Why not just list the criteria in columns and indicate where models passed/failed?

———————————————

---

## Referee Comment (RC2) · Anonymous Referee #2 · 28 Dec 2019

In the framework of computational scientific reproducibility, this manuscript attempts to explore the status of CMIP5-class climate model accessibility, following a methodology based on web access attempts and emails. The question addressed here is very important, and an appropriate discussion could help the climate model community improving its code sharing practices. Authors show that only very few of these models comply to a full accessibility. Such a result should rock the boat of the climate modelling community and ignite discussion on reproducibility within successive CMIPs. Unfortunately, in its present sate, that is basically the only result this manuscript provides, and the text suffers from a lack of discussion and perspective, together with far too many general and un- or ill-referenced statements. I think the main results should be followed by a discussion both on the relevancy of code sharing policy for CMIPs, and

more suggestions to improve it. Also an historical perspective of how code availability was considered in the successive CMIP projects would be interesting to evaluate how things are evolving in the climate modelling community.

Many questions come in mind reading the ms, I am not requesting authors to answer them, but I think they could help building a discussion section: I acknowledge authors suggestion to setup "frozen" versions of the codes, but in more details, what could be suggested, at the international level, to improve sharing policy of the codes/experiments ? What has been done in the past? Why did it not work until now ? Do we have information regarding the ongoing CMIP6 experiments ? What are the specificities of climate model intercomparison projects when compared to other massive modelling works ? Making codes available is one thing, but does it make sense to make a million-line-ish code available without any support ? Can modelling group follow a standard for documenting the setup of a typical CMIP experiment ? Is it possible ? How the ultimate goal of reproducibility could be reached for CMIP models ? On what machine ? Should the community think of compiling/running all the CMIP models on one single machine ? What are the limits of reproducibility in that case ? What was the situation for previous CMIPs, when there were less models involved ? How thoughts on code accessibility did evolve ?

On the methods: Although I am not an expert in surveying methods, I found it puzzling not to have more details about the emailing methodology, i.e. who was contacted in the different modelling group: engineers, researchers ? Climate/ earth system models are massive codes, developed by many people. Did the authors contact, for each model, responsible for each compartments (vegetation, atmosphere, ocean, etc.) or did they just take one contact from each model web page ?

On the results : I did not find the geographical (fig 1) approach relevant. What conclusion can be drawn from that ? Although it is more complicated, I think having a licensing history of each model would be more relevant to connect to their accessibility.

[Figure]

Specific comments :

P1,l7: "There are other reasons that justify the need for access to the codes of climate models used in scientific research. One of the most important is to prevent the loss of knowledge on the cycles of development of these models. Some of them nowadays rely on 'legacy' code that was written up to five decades ago, and new developers must understand why some decisions on implementation were undertaken so long ago. Âż Although I am convinced by the need of improving code sharing policy, I am not sure that it will help reducing loss of knowledge. From my experience it seems that many steps to improve a climate model code are either recorded internally, i.e. within the institutes documents, or through successive publications. I don't see how code sharing will improve that.

P1 L13: "The complexity of the problem, where in some cases scientists may be un-aware of some of the determinants, or may make subjective judgements that have little to do with the most appropriate from a scientific point of view (Joppa et al., 2013), or may even fail to make the correct assessment, makes it necessary to consider a range of issues (Añel, 2017), including legal aspects.".

I must confess I don't understand this sentence.

P2l13 "It could be said that adequate sharing and documentation is not necessary if the code used in the models includes appropriate comments, but it is generally the case that climate models do not comply with what would be the ideal level of programming practice." That is a really strong statement that should be underpinned by appropriate reference.

"It is widely acknowledged that some scientists are reluctant to share code because of the perceived potential damage to their reputations." I am really surprised by this statement. Is it supported by any survey ?

"Given that many scientists have no formal training as programmers, it may be presumed that they consider that their code may not comply with the standards of excellence that they usually pursue in their main fields of knowledge. Indeed, it has been clearly documented that some climate scientists acknowledge that imperfections in climate models exist, and they simply address them through continuous improvement without paying too much attention to the normal techniques of software development (Easterbrook and Johns, 2009). Nevertheless, all scientists must believe that their code is good enough (Barnes, 20 2010) and that there are thus no reasons not to publish it (LeVeque, 2013)." This paragraph, as the previous sentence, suggest climate scientists, aware of their code imperfections, would be reluctant to share it. It must be supported by a reference or a survey, if not it is just a feeling.

"Barriers to code-sharing through licensing, imposed by e.g., government bodies, cannot be an excuse and when contributing to scientific studies and international efforts where collaboration and trust are critical, such practice is not acceptable." This sentence is more an open-ed-like statement that what is expected in a scientific journal. Questioning licensing is appreciable but it should be made in a more rigorous way.

"For cases where we obtained the code of a given model, we were not provided with a reason for the license behind it. In fact, in some cases despite getting the code we did not see a license explaining clearly the terms of use." Indeed it would have been crucial to obtain, for every model and every component, the license terms used. That would have helped a lot to discuss accessibility.

---

## Author Comment (AC1) · 19 Feb 2020

**Reply to reviewers**

We thank the comments by all the reviewers. We have prepared a new version of our paper addressing the concerns pointed out to the best possible. These improvements include some extra information in new sentences, some rewriting and a few further references. Point by point replies follow next.

**Reply to 'SC1: 'Comments of paper contehnt', Richard Neale, 08 Sep 2019'**

First of all, we thank Dr Neale for taking the time to read our work and provide feedback on it. We have made our best to address the concerns pointed out. Dr Neale states:

| |
|---|
| **I just wanted to comment on the aim and transparency of the work in the paper.The topic of this 'study' certainly has merit as modeling centers are trying to improve their transparency and availability of model code, but it is an ongoing process in the face of certain restrictions that were noted.** |
| We fully agree on this point with Dr Neale and, we are pleased to know that it has become clear in our work. |

| |
|---|
| **I have significant concerns though, regarding its approach and methods and inappropriate speculation. One thing that stood out for me as complete speculation is the line "It is widely acknowledged that some scientists are reluctant to share code because of the perceived potential damage to their reputations". This is quite inappropriate for an academic journal, and I really do not understand what is meant by this. Being pretty intensively involved in CMIP efforts for many years, I have never come across code release reluctance related to 'reputation'.** |
| We fully agree that inappropriate speculation is not right for an academic journal. Unfortunately, our experience does not agree with this view by Dr Neale. We are not unaware of the evaluation processes of climate models. The first author of the work has participated in WMO reports on models, their use for data validation, has published several papers on model evaluation, participated in several workshops on the subject and has been editor and reviewer of multiple papers on climate models. Concerns on the perceived lack of quality of the code of models often appear in the comments of model developers and colleagues during workshops. The problem of the 'reputation' is linked to the quality of the software. Some people think that low-quality software could damage their reputation. |
| But beyond our perception, we present evidence that supports informed speculation. Indeed, as we state and show in our submitted work and cited in the text (García-Rodríguez et al.), climate models are inadequate in terms of programming practices (the paper (currently under review) where this evaluation is performed is accessible from this link: http://fortrananalyser.ephyslab.uvigo.es/). This issue happens in some cases to a level that could be embarrassing for a project of an undergraduate student in computer sciences. Also, Barnes (2010) pointed out this issue and Wieters and Fritzsch (2018) – now cited in the text - makes a similar statement. |
| However, following the concern raised by Dr Neale and agreeing that the sentence in its current form is probably not enough supported by objective evidence, we have rewritten it as follows: |
| *'It is possible to speculate that some scientists could be reluctant to share code because of perceived potential damage to their reputations (Barnes, 2010) because of its quality, perhaps related to a lack of adequate education in computer programming (Merali, 2010). This issue with the quality of the code has been raised for the case of climate models (García-Rodríguez et al., 2020).'.* |

> **Furthermore, we simply have no idea how rigorous the investigation is. It quoted a whole raft of comments from various staff at institutions that cannot be attributable to anyone. We have no idea whether they spoke to junio scientist or a senior director.**

We do not understand this claim by Dr Neale. We consider that subsection 2.1, Tables 1 and 2 and Appendix A provide plenty of information on the rigour of our procedures. We have tried to get the access to the code of the different models: use of the CMIP5 webpage, internet search and contact by email in two different languages, several tries to contact by email plus further discussion in a workshop.

The point on attribution is clear: First, we do not decide who reply to the emails or the contact point for each developing model group. How to contact is usually included in the CMIP5 web page or the webpages of the models. We can not decide if it is a junior computer scientist or a senior director. It merely is the contact point that the developing centre/team has chosen. Secondly, who is the contact is irrelevant here, and it is not our fault if it fails to give a reply or the content of the response that we receive. It is the 'official' contact point for the model developing team, and therefore it must be assumed that the reply is 'official'.

A final issue is about the possibility of checking the replies by the different research centres or developing teams. This issue was already pointed out by the editor in a previous version of this manuscript. Of course, we can not publish the replies that we received, because they are subject to communications privacy. However, as in any review process, we offer the possibility to the editor of checking them if he considers it necessary.

> **And it is worth pointing out that a little deception was used in the attribution to a researcher as a PhD student in their emails.**

Unfortunately, again, we do not clearly understand this statement by Dr Neale. Does he suggests that we have tried to cheat or mislead the developing teams on our intentions?. It could not be farther from the truth. Mr Michael García Rodríguez is a PhD student at our university. We do not understand what is wrong with it. Indeed, as we pointed it out on page 3, line 12 in the previous version of the manuscript, the code of the models must be available to everybody, not only researchers.

> **Don't get me wrong, easily accessible access to code is important in the whole climate change and climate modeling sphere, but the authors need to rethink this approach quite a bit. Of course I will let the handling editor and reviewers determine whether they agree with me :) Thanks Rich Neale**

We thank all the comments by Dr Neale that we consider constructive. However, concerning this comment, we do not understand what he means by 'need to rethink this approach'. Probably this happens because the word 'approach' is not specific enough. Does it refer to how we have tried to get the code? We will be happy to address any concern relative to our study if it is clarified.

**Reply to "Anonymous Referee #1"**

**Review of "Current status on the need for improved accessibility to climate change models" by Juan A. Añel, Michael García-Rodríguez, and Javier Rodeiro.**

**The manuscript looks into the availability of the source code of models which contributed to CMIP5. The authors basically use a three-step approach to get access to the code (direct download, "anonymous" email and email stating their research). Their results show that more than half of the model source code is not made made available after step three. In addition, they discuss the documentation quality and licensing issues. I find the topic of the manuscript to be highly relevant and the results presented by the authors raise a crucial issue which is of high importance for the climate model community. However, I find the manuscript to have several weaknesses which should be addressed before a possible publication in Geoscientific Model Development as outlined in the comments below.**

We thank the Referee#1 for the supportive comments on our manuscript. Next, we reply point by point to the concerns raised.

**General comments:#1**

**The manuscript contains several generalized statements which I find to need further support by (scientific) literature or the results. I've addressed them also in the specific comments but in general I refer to statements like the following: "...it is generally the case that climate models do not comply with what would be the ideal level of programming practice." (page 1, line 11)**

This sentence is actually on page 2, lines 12-13. We consider that our statement is supported enough by the results of the work cited in the text: García-Rodriguez et al. In this work, climate models are one of the case studies used to prove the usefulness of a Fortran code static analysis tool. It is available through the link http://fortrananalyser.ephyslab.uvigo.es/, something that we make clear now, including the link in the reference. Also, a careful reading of Easterbrook (2014), cited in the sentence that follows the one pointed out by the reviewer, lets to reach a similar conclusion. Moreover, Wieters and Fritzsch (2018) – now cited in the text – pointed out the same conclusion regarding software engineering practice in the field of climate modelling.
Later, in the Conclusions section, we cite now Merali (2010), that includes some further discussion on the topic.

**"...the incidence of comments throughout the code is very low" (p1, l13)**

Again, this is supported by the results in García-Rodríguez et al. We acknowledge that in the previous version of the manuscript, the reviewers could have problems to get access to this information. However, now we link this study submitted for publication.

**"It is widely acknowledged that some scientists are reluctant to share code because of the perceived potential damage to their reputations." (p4, l14)**

Dr Neale raised concerns about this sentence too in a short-comment. We have modified the sentence to make it less speculative, and now we cite the work by Merali (2010) to provide additional evidence and support. The new sentence reads:
'*It is possible to speculate that some scientists could be reluctant to share code because of perceived potential damage to their reputations (Barnes, 2010) because of its quality, perhaps related to a lack of adequate education in computer programming (Merali, 2010). This issue with the quality of the code has been raised for the case of climate models (García-Rodríguez et al., 2020).*'.

**2 I find the discussion about the need to publish model code in the introduction too one-sided. While the authors give several good reasons why code should be made publicly available, there might be equally valid counter-arguments. It would be good to discuss some of them as well and possibly offer solutions. Some things that come to mind:**
- National or institutional copyright that prevents authors from publishing code
- Dependencies of the model on third party code that is under copyright
- A lack of funding to set up and maintain a public code repository
- Fear that ones property rights might be violated if the code is freely available in the web.

This might be particularly true for newer models. An argument could be made that a group developing a model has the right to also publish the results produced with this model (they might even be required to do so by their funding agency).

The referee makes some good points here but also makes some mistakes, as the incorrect use of 'copyright' to refer to very different legal aspects that could involve intellectual property, redistribution, reuse and modification of software. These comments are in line with usual concerns by many colleagues, and we have made an effort to include them in the text while making clear the most critical issues.

That said, we do not think that the discussion of the points suggested by the referee corresponds to the Introduction, but the Conclusions section. We address the concerns here and in some extra text that, therefore, we have added in the Conclusion.

First of all, respectfully, we do not consider that our discussion in the Introduction is one-sided. It is a mistake to think that there are two equally valid sides here. Someway it is the same fallacy as considering that negationist arguments on anthropogenic climate change deserve to be highlighted to the same level that the scientific consensus on it. The only valid side is complying with the scientific method. If we do not apply the scientific method, what is produced can not be considered science. It is clear that if the access to the code of the models is not provided, we do not comply with the scientific method, as the primary tool used to produce the results is a black box. This problem has been acknowledged many times in the literature. It inspires the editorial policies of many journals. Stodden et al. (2013) (already cited in the previous version of the manuscript) discuss more in deep this issue. However, to better support it, and having into account that it is a pivotal issue for the understanding of the goal of our study, we have modified the first sentence in the paper and added two additional references. The new sentence reads:

"*Reproducibility of results is essential to comply with the scientific method when performing research. This has extraordinary implications in the field of earth system models (Añel, 2019; Gramelsgerber, 2020)*"

The problems with computational reproducibility are evident in some of the works that we already cited in the previous version of the manuscript: Allison et al., 2018; Stodden et al., 2018.

To include the so-called counter-arguments, we have added in the discussion a few paragraphs that read:

"*Some scientists or model development groups could be worried because of issues such as legal restrictions (national or institutional) that prevent from publishing code. Also, because of potential dependencies of the model on third party proprietary software, lack of funding to maintain a public repository or violation of property rights. For all these cases, there is a clear response or solution.*

*In the first case, if it is not possible to make available the code, then any result obtained with such a model should not be accepted as scientifically valid because it is impossible to verify the findings. We acknowledge that this is the case for several models widely used in scientific research nowadays and must be solved by modification of the legal precepts applied to them. Consequently, those working with such models should look for a change in the legal terms so that the model complies with the scientific method.*

*If a part of the model depends on proprietary software, then it harms the possibility to distribute the whole model. Therefore again, the model does not comply with the scientific method. In this case, a good option can be to substitute the part of the model that is proprietary software with one that is free software.*

*Lack of funding to maintain a software repository can not be considered a real problem. There are many options available to host the code. For example, Zenodo is free, widely adopted and assures hosting at least for the next twenty years.*

**#3 The way the authors tried to establish contact is not clearly enough documented and needs clarification. As it is, it seems quite subjective to me. In particular:**
**- How did the authors search for contact information? They mention that they searched the internet.**

To try to clarify the issue of how we search for contact information, we have modified the text in the manuscript. Now it reads:

*'This procedure included the first step using the web addresses for code downloading indicated on the CMIP5 webpage (\url{https://pcmdi.llnl.gov/?cmip5/}). When the code was not directly available to be downloaded, we looked for contact details (emails, on-line formularies, etc.) in the webpage. In some cases, there was an email address to contact. However, in other cases following the information in the CMIP5 website was not enough. In such cases, we searched through the internet using a search engine. We looked for institutional web pages, intending to find open repositories for the corresponding model.'*

**But how easy was it to find the information and at what point (after how much time) did the authors give up? One could for example go as far as looking for publications by the same group investigating the desired model and writing the authors. I was quickly able to find contact information for all five model centres the authors list as "No email or contact phone is available" (see specific comments below).**

It is someway hard to quantify 'how easy' it was to find the information. In some cases, it was direct from the webpage of the CMIP5, for others, as we already explained in the previous version, we had to look for it using a web search actively. However, we do not consider that 'pursuing' the developing teams by looking for contact emails in scientific papers should be assumed as a way of doing it. First of all, many papers continue to be accessible only through paywalls (not open-access) and therefore the information there is not available for most of the people. Secondly, looking for papers, identifying the relevant ones and the appropriate person to contact from an author list requires a knowledge of the modelling groups that only a handful of experts in the field have. And again, availability of the code should be judged having into account that the code of the models must be available to everybody, without being an expert in climate modelling. This is why we did not consider looking through the scientific literature as an appropriate way to assess the availability of the code of the models. In the same vein, to use the metadata in the output NetCDF files of the models, as the referee mentions in the specific comments, is only available to a very specialized small group of people. However, we have introduced a new paragraph in subsection 2.1 to make clear these issues.

*'It could be possible to look for additional contact information in the published scientific literature. However, many papers continue to be accessible only through paywalls (they are not open-access) and therefore they are not available for most of the people. Moreover, identifying the relevant person to contact from an author list requires a knowledge of the modelling groups that only a handful of experts in the field have. Also, additional contact information is available for five models from metadata included in the NetCDF files containing the results of the simulations in the CMIP5 repository. However, the ability to find and manage such data has computational requirements and needs of a knowledge that is beyond what could be considered reasonable for the general public, including part of the scientific community.'*

**The two mails in A1 and A2 are identical. Is that a mistake or did the authors just send the same mail again? In this case they could consider deleting A2.**

We send the same email again. It was already exposed in the previous version of the manuscript just before the content of the email. We assumed that in some cases the contact person could be busy, that the email could get simply buried in the inbox or may be filtered as spam. This is the reason why we did a second try with the same email. As suggested by the referee, we have removed now the second email from the text of the manuscript. To clarify it, we have added in the text a sentence in subsection 2.1 that reads:

"*A second email, equal to the first one, was sent to insist on our request. We intended to minimize the possibility of not getting a reply because of reasons such as the contact person being too busy at the moment of receiving the first email, it was unnoticed or filtered as spam. We waited for a reply for three weeks after sending the first email before sending the second. Finally, three weeks after the second email, we sent a final email, where we identified...*"

**In section 2.1 the authors mention that in "the final email" they identified themselves. After how many mails of type one was that? The "final email" should also be in the appendix.**

This was the third email, as it is listed in Appendix A. It has been already clarified in reply to the previous comment. The 'final email' was in the prior version of the manuscript but as 'Third email'. After the additional explanations included in the text, this is now clearer.

**In table1 only email 1 & 2 are listed. Are these two columns indeed refering to the identical emails from A1 & A2? If yes I'm missing the column for the final email.**

For the final third email, we did not get a single reply. Therefore we did not include a column for it. We have clarified it in the caption of the table.

**The authors mention one specific model for which they got access after direct contact to the developers at a conference. Was this approach tried for all model centres which did not reply to the emails or was it based on a coincidence? If it was only done in this one case, one could argue that this partly jeopardizes the objectivity of the approach.**

We did not make an active approach to get access during conferences. As the referee points out, this would jeopardize the approach. Moreover, it would not be fair for the evaluation, as the general public does not attend scientific conferences. What happened was that the corresponding author of this work was approached by one of the members of the developing team at NASA. Then this person put us into contact with the same colleague that we had already tried to contact before (the contact point for the model) and they sent us the code.

We have modified the relevant sentence in subsection 2.1 to make clear that we did not take an active role to contact groups during conferences and to provide additional details. Now the sentence reads:

"*For the NASA-GMAO model, we failed to get a reply from contact via email. However, we were approached by a member of the institution after a presentation during a conference. After discussing it, the development team granted us access to the model.*"

Moreover, although finally we took the opportunity to analyze the model, NASA-GMAO failed in the availability of the code. This issue is recorded in Table 1, failing to get a third star because of it.

**#4 The manuscript would benefit from proof-reading by a native speaker. There area number of very long and somewhat convoluted sentences. This sometimes makes it hard to follow the authors point as I mention at several occasions in my specific comments.**

We thank this comment. Indeed, our submitted manuscript had already been edited by a professional service, as all of our papers. However, there is always a margin for improvement. The new version of the manuscript has been rechecked.

**Specific comments:**

**Title: Maybe change to "model code" in order to make clear that this is about the source code and not about output? Also I believe that it would be more appropriate to call them "climate models" instead of "climate change models" because they are used to investigate the climate system in general not only climate change.**

We agree with the referee about the title, and we have modified it accordingly. Thank you for pointing it out. However, we have prefered to maintain 'climate change models' in the text. Indeed the models are 'climate models', but our testbed was the CMIP5, therefore models specifically intended to inform on climate change and to serve to the IPCC reports. Because of this reason why we think that 'climate change models' is more accurate here.

**Abstract: "models from the Climate Model Intercomparison Project" Add "fifth"**

Done.

**Page 1, line 13-16: This is a very long sentence and I'm not quite sure what the authors try to say here. Is this addressing the issue of reproducability in general? E.g., subjective judgments (as long as they are properly documented) to not hinder reproducability. So maybe the authors try to say something else? Please clarify, since this seems to be an important point.**

We have rewritten the sentence to try to make it clearer:

"CSR is a problem of high complexity. In some cases, scientists may be unaware of some of the issues that have significant impacts on it, reaching the wrong conclusion that an experiment complies with CSR, when it is not the case, as exposed in Añel (2017). Also, a researcher could decide to use a model based on judgements that have little to do with the most appropriate from a scientific point of view, such as complying with the scientific method (Joppa et al., 2013). All this makes it necessary to consider a range of issues to comply with CSR in the process of design and use of models (Añel, 2017), and climate change models in particular. Some of them are legal aspects of software distribution and intellectual property, usually unfamiliar for researchers."

**P1, l16: replace "matters" by "CSR"**

Done

**P1, l19: It would be convenient for the reader to have the (most important) recommendations listed here, instead of only quoting Wilson et al. 2017.**

Wilson et al. (2017) include many recommendations that affect different issues. Now we list in the text some of them that we consider representative and relevant for our work.

**P2, l11: "It could be said that adequate sharing and documentation is not necessary if the code used in the models includes appropriate comments," I'm not sure what the authors try to say here. Why would commenting the code make publishing unnecessary?Also I find the statement that commenting code can replace a proper documentation problematic.**

We fully agree with the referee. Indeed, what we want to say here is that both right code and documentation are needed. We have made our statement clearer, and we cite a paper (in press) by Pascoe et al. that makes some good points on the need for documentation in the CMIP. Now the sentence reads:

"*It can be argued that adequate documentation of the code and the model is not necessary to prevent a potential loss of knowledge if the code used in the models includes appropriate comments. But, indeed, this is not the case of the models contributing to the Fifth Climate Model Intercomparison Project (CMIP5). CMIP models are sophisticated software projects, and they need full documentation of the experiments (Pascoe et al., 2020). Moreover, it is generally the case that climate models do not comply with what would be the ideal level of programming practice.*'.

**P2, l12-15: "...but it is generally the case that climate models do not comply with what would be the ideal level of programming practice." I'd like to see some evidence to support this statement.**

As we have said before, we now include a link to our submitted work García-Rodríguez et al. (2020), a paper on a software tool (FortranAnalyser) that we have designed to evaluate the quality of static code in Fortran. There, it is possible to check how some CMIP5 models perform in terms of programming practice.

**P2, l13-15: "Indeed, the incidence of comments throughout the code is very low, and programmers have tended to perform very badly in this regard in particular (García-Rodríguez et al., 2019)" What code to the authors refer to here? What does low number of comments mean? The citation (García-Rodríguez et al., 2019) is listed as submitted, it should be provided.**

We refer to the code of the CMIP5 models, and it is now clarified in the text. García-Rodríguez et al. (2020) is now provided.

**P2, l21: CMIP 5 I assume?**

Right. We have corrected it.

**P2, l30 I wouldn't call two different emails a "variety of different approaches".**

We have rewritten the sentence to make it more specific:
'*...and contacted research groups where necessary using email, without disclosure of ourselves as climate scientists to full explanation of our interest in studying the code. These approaches are detailed in the following sections.*'

**P3, l3: add "as a first step..."**

Unfortunately, we do not understand this comment by the referee. In P3, l2 it is already said '...a first step...'. If the referee can clarify it, we are happy of addressing the issue in a next round of the review process.

**p3, l3: Could the authors provide a link to the CMIP5 webpage they refer to here?**

Done.

**P3, l5: Why English and French? Several models are developed, e.g., in China it seems to me that for a systematic and objective methodology it should be either only English (the language used in the vast majority of climate model related publications) or all respective native languages of the model centres.**

We sent first the emails in English to all the research centres. From one of the research centres, the IPSL, we did not receive a reply. As one of the coauthors of this work is a French native speaker, we decided to send the second email in French, just in case we had a better chance of getting a reply. We were successful, and we got access to the code of the IPSL model. We have rewritten the relevant part of the text to make clear that sending out emails in French was not the rule. Now the text reads:

*"Still, in others, we had to proceed by making contact with development teams at different levels (emails (see Appendix) (with follow-up emails two weeks after the first contact). To contact the IPSL team after sending the email in English and failling to get a reply we sent the second email in French)"*

**P3, l11-12: "to check whether after it had become obvious that the models were not available easily, institutions and researchers would then share them with someone from the general public." To really represent the general public it would have been better to use a not-university related mail address I assume. Maybe change to something like"with someone from outside the community"?**

Done.

**P3, l18: Why did it take several months? The authors state above that they send an initial email with a follow-up two weeks later. Was it the process of actually getting the code after establishing contact that took so long?**

We have opted to write 'several months' because it was the time that it took from the beginning of the experiment (checking the CMIP5 webpage) to obtaining the last model that we got after contact during a conference, the one contributed by NASA-GMAO. We consider this a minor issue and according to what happened. Therefore, we have not modified it.

**P3, l18: To I assume correctly that "10 out of 26 models" referrers to "models from 10 out of 26 institutions" and "27 out of 61" referrers to multiple models from the same institution (such as GCM and ESM versions)?**

We think so. However, the comment by the referee is not too clear here. We feel that it is evident in the text: There are 26 models, but some of them (as listed in Table 2) have contributed different versions, with a total of 61.

**P3, l23: The percentages refer to the numbers in the first line of the paragraph, I assume. The "they" seems to indicate that only USA, Germany, and Norway are meant.**

Correct. We have modified the sentence to clarify it: «Together, these three countries represent...»

**P3, l24-26: "This analysis is relevant, because in some cases the decision on whether to share the code of the models could have been due to national or regional regulations on software copyright, intellectual property, etc." It would be interesting to detail this point further. Did the authors get concrete answers citing (national) copyright law as response? Did they check the copyright law in countries where they did not get access to any models?**

No, we did not get replies citing national laws. When we got a response with the reason to deny access (five cases), it was claimed that access to the code is only granted to members of the development team. Additionally, in two cases, the reply mentioned that moreover, it would be unfeasible because of the size of the model and that it would involve too much work. This information was already included in the previous version of the manuscript in Tables 1 and 2. Checking the copyright laws that apply to different regions (and probably institutions) is out of the scope of this work. Moreover, in some cases, this could respond to regulations established in confidential information of national laboratories, etc. In this way, it is hard to know if it is possible to perform a proper analysis of this matter.

We agree that our statement is slightly speculative, and therefore, we have modified the sentence to make it clear. Also, we introduce some further discussion:

"*This analysis is relevant. We can speculate that in some cases, the decision on whether to share the code of the models, could have been influenced by national or regional regulations on software copyright, intellectual property, etc.* For example, it is well known that the law in the USA, where for example software can be patented, enables the possibility to enforce a greater level of restrictions to software sharing and distribution than in the EU (van Wendel de Joode et al., 2002; EPO, 2016). However, it is the case that we have been able to get the code for 6 of the 7 models contributed by research centres from the USA and however, for the EU we have only got 3 in 7. The fact that the models developed in the USA can count with the participation of federal employees could partially explain this result. Under the USA copyright law (U.S. Code, 1976), all the work produced by federal employees is in the public domain. Although they are not the same thing, the public domain could be considered closer to openness that lack of licensing of the models. For the case of Norway, we can speculate that the fact that NorESM has been developed using core parts of CESM1 (Knutti et al., 2013) could have facilitated openness of the code through the inheritance of licenses and copyright. In the same way, not sharing the code of the models could be due to inheritance reasons."

**P3, l30: "in six cases" & "in five cases" Table 1 only lists 5 cases with "No email or contact phone is available." and only 4 cases with "No answer".**

We thank the referee for the careful reading and checking of the manuscript that made possible to find this subtle mistake. We have corrected it.

**P4, l3-5: "We considered the level of requirements introduced by the GPLv3 license (https://www.gnu.org/licenses/gpl-3.0.en.html) as the ideal case, or a license under which the model can be shared, modified and used without restriction." This seems to be contradictory. Is the GPL the ideal case or "a license under which the model can be shared, modified and used without restriction"? Because to my knowledge the GPL has very strict requirements that need to be fulfilled to share, modify and use code which is licensed under it (such as: disclosure of source, stating of all changes, further publication only under the same license).**

The GPL is one of the two licenses recommended in the GMD Editorial (GMD, 2019) that we already cited in the manuscript. Exactly the GPL is one of the few licenses (beyond some of the considered as 'GPL compatible') that lets to use the software obtained without limitations. Some people argue that the GPL imposes limitations. However, this is only the point of view of a developer that wants to restrict sharing of the software. The GPL is the kind of license that better complies with the spirit of science and as Morin et al. (2012) say '*assure the benefits and openness of FOSS in all future derivatives of your work*'. Therefore, to make the reasons for choosing the GPLv3 as the ideal license, we have rewritten the sentence. Now it reads:

"*This is in line with the recent updates to the policy on code availability published by Geoscientific Model Development (GMD Executive Editors, 2019). Moreover, it has been argued that it is the license that better fits to scientific projects to assure the benefits and openness of software (Morin et al., 2012).*"

**P4, l14-15: "It is widely acknowledged that some scientists are reluctant to share code because of the perceived potential damage to their reputations." Can the authors provide some evidence for this statement?**

This issue has been addressed. Please, check previous replies.

**P4, l20-23: "Barriers to code-sharing through licensing, imposed by e.g., government bodies, cannot be an excuse and when contributing to scientific studies and international efforts where collaboration and trust are critical, such practice is not acceptable." It seems to me that following the laws of ones country (even when they hinder research and collaboration) is indeed a completely valid excuse for not sharing code.**

We agree that following the laws is a valid excuse for not sharing code. But not doing it means that the results produced are not legitimate science. The scientific method is clear about this: Results have to be reproducible to be considered scientifically valid.

At this point, what should be clear is that the research community should pressure a government or institution to get a change of the laws or licenses that applies to the software developed under their legal framework. In this way, their results could be acknowledged as valid by the research community.

**P4, l22-24: "For cases where we obtained the code of a given model, we were not provided with a reason for the license behind it. In fact, in some cases despite getting the code we did not see a license explaining clearly the terms of use." It would be interesting indeed to know the rationale behind different licenses (or for the absence of a license), did the authors inquire about this at the groups which provided code?**

We asked for the reasons for not sharing the code in the third email. However, as we explain in the manuscript, we did not get an answer to these emails.

The rationale for the different licenses applied probably embrace from only the license that the first group of developers choose to an imposition by a research institute. However, this is a subject to be discussed more from the point of view of law and psychology. We consider it interesting, but out of the scope of this work.

**P4, l31-32: "we encourage all model developers to improve the availability of the codes of climate models and their CSR practices." As I've mentioned before this might not be only the developers responsibility but also includes institutions, funding agencies and even country copyright laws.**

The referee is right. However, we encourage the developers to make clear to the relevant bodies or agencies that it is not acceptable to comply with the scientific method.

**P4, l33: "which is in some cases very poor (García-Rodríguez et al., 2019)" Again, please provide this paper as it is not yet available.**

Please, see replies above.

**Figure 1a: Maybe delete the axis? (or fix the y-axis, which should run from -90 to 90 I assume?) The percentages are not per country as stated by the caption but by continent I assume? Could the authors find a way to explicitly show that there are now model centres in South America and Africa?**

**Figure 1: I personally would find it more helpful to see absolute numbers of provided/not provided models instead of percentages.**

We have modified the figure according to the suggestions by the referee.

**Table 1: I checked our CMIP5 archive and found all of the contact information flagged as missing in the metadata of the output netCDF files for the respective models. I understand that at this point it is probably too late to include them in the study, yet I'm copying them in here in case they are helpful to the authors.**

**BCC: "contact = "Dr. Tongwen Wu (twwu@cma.gov.cn)""CSIRO-Mk3.6.0:"contact="Projectleaders:StephenJeffrey(Stephen.Jeffrey@qld.gov.au)&LeonRotstayn(Leon.Rotstayn@csiro.au).Project team:Mark Collier (Mark.Collier@csiro.au:diagnostics & post-processing),StaceyDravitzki(Stacey.Dravitzki@csiro.au:post-processing),CarloHamalainen(Carlo.Hamalainen@qld.gov.au:post-processing),SteveJeffrey(Stephen.Jeffrey@qld.gov.au:modeling&post-processing),ChrisMoeseneder(Chris.Moeseneder@csiro.au:post-processing),LeonRot-stayn (Leon.Rotstayn@csiro.au:modeling & atmos.physics), Jozef Syktus(Jozef.Syktus@qld.gov.au:modelevaluation),KennethWong(Ken-neth.Wong@qld.gov.au:data management), Contributors:Martin Dix (Mar-tin.Dix@csiro.au: tech. support), Hal Gordon (Hal.Gordon@csiro.au: atmos. dy-namics), Eva Kowalczyk (Eva.Kowalczyk@csiro.au: land-surface), Siobhan O\'Farrell(Siobhan.OFarrell@csiro.au: ocean & sea-ice)""INM-CM4: "contact = "Evgeny Volodin, volodin@inm.ras.ru,INM RAS, Gubkina 8,Moscow, 119333 Russia,+7-495-9383904""LASG-IAP: ":contact = "Dr. Tianjun Zhou(zhoutj@lasg.iap.ac.cn)" MRI: "contact = "Seiji Yukimoto (yukimoto@mri-jma.go.jp)""**

We thank the referee for taking the time to look for the information in the NetCDF files. However, as we have discussed previously, dealing with this kind of files is challenging for most of the scientific community and the general public. Therefore, we have not considered in our study NetCDF files as a valid form of providing contact to get the code of the models.

**Table 3: I find the star-rating system slightly in-transparent. Why not just list the criteria in columns and indicate where models passed/failed?**

This suggestion looks reasonable. Admittedly, we did a try with this, with a table including four additional columns for direct download from the CMIP5 webpage, reply to the first email, reply to the second, GPLv3 compatible license, etc. However, the result included leaving blank boxes when emails were not necessary, etc. Moreover, all this information is included in the previous tables and the text. In the end, we did not feel that the result was better than the current version only with the stars. We have decided simply to modify the caption of the table to clarify that the maximum value is three filled stars, something that was not clear before.

**Reply to Anonymous Referee #2**

We thank the referee for the insight provided, and the relevant questions pointed out.

> **In the framework of computational scientific reproducibility, this manuscript attempts to explore the status of CMIP5-class climate model accessibility, following a methodology based on web access attempts and emails. The question addressed here is very important, and an appropriate discussion could help the climate model community improving its code sharing practices. Authors show that only very few of these models comply to a full accessibility. Such a result should rock the boat of the climate modelling community and ignite discussion on reproducibility within successive CMIPs. Unfortunately, in its present sate, that is basically the only result this manuscript provides, and the text suffers from a lack of discussion and perspective, together with far too many general and un- or ill-referenced statements. I think the main results should be followed by a discussion both on the relevancy of code sharing policy for CMIPs, an more suggestions to improve it.**

Admittedly, we like the idea of including additional discussion on the topic in the text. We would like to have done it in the previous version of the manuscript. However, this discussion could lead to too much speculation. We are aware (we have discussed this with members of some development teams) that some colleagues are someway reluctant to change how they develop the models. We prefer to avoid strong statements in the manuscript. Already some of them are considered by the referees pretty speculative in the previous version of our paper.

To try to shed some light on how the CMIP can improve this situation, we have included some new suggestions in the text. We have rewritten the last paragraphs of the Conclusions. Among others, we suggest a potential integration of the evaluation of the code of the models in the ESMValTool. Also, now we expose that there are discrepancies between outputs of CMIP5 models, and we discuss how sharing the code would help to address them. This has already been proved in other fields of research and was cited in the previous version of the manuscript (Boulanger, 2005).

> **Also an historical perspective of how code availability was considered in the successive CMIP projects would be interesting to evaluate how things are evolving in the climate modelling community.**

Again, this issue raised by the referee is fascinating. However, we think that this is a specific research topic, and therefore it should be the result of future work. Indeed, we aim to do it.

> **Many questions come in mind reading the ms, I am not requesting authors to answer them, but I think they could help building a discussion section: I acknowledge authors suggestion to setup "frozen" versions of the codes, but in more details,what could be suggested, at the international level, to improve sharing policy of the codes/experiments? What has been done in the past? Why did it not work until now?**

As we exposed in the previous version of the manuscript and we have tried to make more evident now, it should be done in the same way that it is done for data output. We suggest that each set of outputs from a model has linked the code of the model used to produce it. Other policies could include proper static analysis of the code, integrating it in the ESMValTool as it is suggested.

> **Do we have information regarding the ongoing CMIP6 experiments?**

No, we do not have it. At the moment, the CMIP6 webpage only contains some model output data and metadata for the models. This includes email addresses for those involved in the development of each model. However, nothing is said about the code of the models. Also, additionally, ES-DOC includes part of the information that was available directly in the webpage for the CMIP5 (webpage of the development team, the model or institution, associated research paper published and an email for contact). Also, it includes a license for the data but not for the model.
Given that, and having into account that CMIP6 is under development yet, we have decided not to include in our paper a discussion on it, as it would be biased.

**What are the specificities of climate model intercomparison projects when compared to other massive modelling works?**

Admittedly we have not found information about this. Moreover, although we are aware of comparable efforts in others field of research (genomics, particle physics), we do not feel prepared to make a judgement on this topic and discuss it in the paper. Also, this could distract the attention of the reader to the main point that we make: Sharing the code must be done not only because of practical issues or how the climate modelling community compares to other fields but because of scientific integrity and to comply with the scientific method.

**Making codes available is one thing, but does it make sense to make a million-line-ish code available without any support ?**

Yes, it does. It has been proved that merely making the code available improves code reliability. This was already pointed out when we cited Boulanger (2005). In this new version, we have included some extra support citing a report of the US Department of Defense (2009) that says: "The continuous and broad peer-review enabled by publicly available source code supports software reliability and security efforts through the identification and elimination of defects that might otherwise go unrecognized by a more limited core development team." (https://dodcio.defense.gov/Portals/0/Documents/FOSS/2009OSS.pdf). Also, Easterbrook (2010) discussed how 'computer-supported collaborative science' is necessary to tackle the challenge of climate change. In this case, this implies having access to the code of the models. We include this statement now in the Discussion of our manuscript, and we cite the work.

**Can modelling group follow a standard for documenting the setup of a typical CMIP experiment ? Is it possible?**

We are not sure if the referee means here to document the code or to document the experiments. The experiments are already documented. Documentation was already addressed for CMIP5 and has been improved for CMIP6 with ES-DOC (now we cite it in the Introduction (Pascoe et al., 2020)). For comments into the code, some clear standards can be followed, and they are part of the metrics used by García-Rodríguez et al. 2020.

**How the ultimate goal of reproducibility could be reached for CMIP models ? On what machine ? Should the community think of compiling/running all the CMIP models on one single machine? What are the limits of reproducibility in that case ?**

Here the referee seems to be confused. It is not the same reproducibility than replicability. The ACM has made it clear and established the differences between them (https://www.acm.org/publications/policies/artifact-review-badging). The question that the referee has pointed here about dependence on machines and limits are related to replicability. It would good to have the highest level of detail possible, such as supercomputer, architecture, operative system, compiler versions and compilation flags. But, in the end, all this means nothing if the source code is not available.

Partially this could be solved using container technologies. In the new version of the manuscript, we have briefly addressed these issues in the Conclusions section and added three further references to support the exposed ideas. It reads:

*"Reproducibility could not be compromised; however, the simple fact that replicability (note that reproducibility and replicability are different concepts (ACM, 2018)) can not be achieved because of the lack of the code of the model is unfortunate. In this way, frozen versions of the models combined with cloud computing solutions and technologies such as containers can be a step forward to achieve full replicability of results in earth system modelling (Perkel, 2019; Añel et al., 2020)"*
* * *
**What was the situation for previous CMIPs, when there were less models involved?**

This question is similar to a previous one by the reviewer about what was done in the past. As we said, it is a complex problem to perform an assessment, and we will try to address it in future work.
* * *
**How thoughts on code accessibility did evolve?**

The answer to these questions is something that could be worthy of addressing, but it would imply a more profound analysis and collaboration from the scientific community. As can be seen from the answers that we have already obtained, to obtain cooperation has already been thought sometimes.

However, again, we take note that this can be of interest, and we thank the comment. This will help to shape our future work.
* * *
**On the methods: Although I am not an expert in surveying methods, I found it puzzling not to have more details about the emailing methodology, i.e. who was contacted in the different modelling group: engineers, researchers ? Climate/ earth system models are massive codes, developed by many people. Did the authors contact, for each model, responsible for each compartments (vegetation, atmosphere, ocean, etc.) or did they just take one contact from each model web page?**

No doubt we have missed some details and explanations on our methodology. We did not check the role of the person that we contacted (when it was necessary to contact somebody). Usually, when the code was not available to be downloaded, it was listed an email contact address for a person or the institution. When it was available, we used the offered contact address, as it was the official contact point listed by the contributing team. In the same vein, when we had to look for the code repository or model development team, we simply looked for the email address listed in the webpage.

We have tried to clarify this point modifying in the Methods section the following sentence:

*"Using a systematic methodology, we attempted to obtain the codes of all the climate models involved. This procedure included the first step using the web addresses for code downloading indicated on the CMIP5 webpage (\url{https://pcmdi.llnl.gov/?cmip5/}). When the code was not directly available to be downloaded, we looked for contact details (emails, on-line formularies, etc.) in the webpage. In some cases, there was an email address to contact."*

**On the results : I did not find the geographical (fig 1) approach relevant. What conclusion can be drawn from that ?**

A straightforward analysis can be done based on the comparison of the application of the law to software between the European Union and the USA. We have included some additional information and references in the text now to make clear the benefits of splitting the analysis by regions and centres:

*"This analysis is relevant. We can speculate that in some cases, the decision on whether to share the code of the models, could have been influenced by national or regional regulations on software copyright, intellectual property, etc. For example, it is well known that the law in the USA, where for example software can be patented, enables the possibility to enforce a greater level of restrictions to software sharing and distribution than in the EU (van Wendel de Joode et al., 2002; EPO, 2016). However, it is the case that we have been able to get the code for 6 of the 7 models contributed by research centres from the USA and however, for the EU we have only got 3 in 7. The fact that the models developed in the USA can count with the participation of federal employees could partially explain this result. Under the USA copyright law (U.S. Code, 1976), all the work produced by federal employees is in the public domain. Although they are not the same thing, the public domain could be considered closer to openness that lack of licensing of the models. For the case of Norway, we can speculate that the fact that NorESM has been developed using core parts of CESM1 (Knutti et al., 2013) could have facilitated openness of the code through the inheritance of licenses and copyright. In the same way, not sharing the code of the models could be due to inheritance reasons."*

**Although it is more complicated, I think having a licensing history of each model would be more relevant to connect to their accessibility.**

We agree that having a licensing history could be interesting, but having into account our knowledge of the field, we speculate that it would be more illustrative than useful. Probably in most of the cases the license was not something relevant when the model began to be developed (mostly for the oldest ones). In the same vein, it is possible that for most of the models when a license was chosen, it was not the result of a profound analysis of the scientific needs and implications, but the result of other issues (requirements from the research centre, etc.). As with the question of how the code has evolved, we take note of this interesting issue, and we will try to address it in subsequent work.

**Specific comments :P1,l7: "There are other reasons that justify the need for access to the codes of climate models used in scientific research. One of the most important is to prevent the loss of knowledge on the cycles of development of these models. Some of them nowadays rely on 'legacy' code that was written up to five decades ago, and new developers must understand why some decisions on implementation were undertaken so long ago. Although I am convinced by the need of improving code sharing policy, I am not sure that it will help reducing loss of knowledge. From my experience it seems that many steps to improve a climate model code are either recorded internally, i.e. within the institutes documents, or through successive publications. I don't see how code sharing will improve that.**

In this regard, the main advantage of code sharing is that it enables the possibility of having more people looking at the code (collective peer-review). Indeed this happens. We agree that it is true that the probability of getting a higher level of lines of code orphaned could not be different between code-shared or not. However, knowledge loss is less probable if more people can access it. We have included two new references in the text (although not in the Introduction) that support this view: DoD CIO (2009) and Easterbrook (2010).

**P1 L13: "The complexity of the problem, where in some cases scientists may be un-aware of some of the determinants, or may make subjective judgements that have little to do with the most appropriate from a scientific point of view (Joppa et al., 2013), or may even fail to make the correct assessment, makes it necessary to consider a range of issues (Añel, 2017), including legal aspects.". I must confess I don't understand this sentence.**

This issue was also pointed out by another referee. We have rewritten the sentence, and we hope that our point is now clearly exposed.

**P2l13 "It could be said that adequate sharing and documentation is not necessary if the code used in the models includes appropriate comments, but it is generally the case that climate models do not comply with what would be the ideal level of programming practice." That is a really strong statement that should be underpinned by appropriate reference.**

We have now included a link to our submitted paper that shows this issue: García-Rodríguez et al. 2020. Indeed, the paper is in collaboration with colleagues from the IPSL and shows as an example how after addressing the most significant problems in the code pointed out by our tool 'FortranAnalyser' the version of the IPSL model for the CMIP6 has significantly improved compared to the CMIP5 version. Moreover, now we cite the work by Wieters and Fritzsch (2018) that makes a similar statement based on their experience.

**"It is widely acknowledged that some scientists are reluctant to share code because of the perceived potential damage to their reputations." I am really surprised by this statement. Is it supported by any survey ?**

We have rewritten the sentence in this version of the manuscript to make more accurate our statement, and we think that now its intended meaning is more precise and supported with cited literature.

**"Given that many scientists have no formal training as programmers, it may be presumed that they consider that their code may not comply with the standards of excellence that they usually pursue in their main fields of knowledge. Indeed, it has been clearly documented that some climate scientists acknowledge that imperfections in climate models exist, and they simply address them through continuous improvement without paying too much attention to the common techniques of software development(Easterbrook and Johns, 2009). Nevertheless, all scientists must believe that their code is good enough (Barnes, 2010) and that there are thus no reasons not to publish it (LeVeque, 2013)." This paragraph, as the previous sentence, suggest climate scientists, aware of their code imperfections, would be reluctant to share it. It must be supported by a reference or a survey, if not it is just a feeling.**

As said before, we have modified the sentence to make our statement more accurate. Also, we have provided new references to support this 'guess', such as Merali (2010) and García-Rodríguez et al. (2020). Moreover, the references previously cited (Easterbrook and Johns, 2009; Barnes, 2010; LeVeque, 2013) already supported with evidence that the issue exists. We agree that it is felling and we acknowledge it. Indeed, the use of the language here exposes it in this way, as the sentence says '*it may be presumed*'.

**"Barriers to code-sharing through licensing, imposed by e.g., government bodies, cannot be an excuse and when contributing to scientific studies and international efforts where collaboration and trust are critical, such practice is not acceptable." This sentence is more an open-ed-like statement that what is expected in a scientific journal. Questioning licensing is appreciable but it should be made in a more rigorous way.**

We do not question licensing. It is not the aim of this work to challenge the existence and application of legal frameworks. What we examine is the possibility of complying with the scientific method when code is not shared. We think that it is clear when we say '*when contributing to scientific studies.*' Now we cite at the end Añel (2019), a work about the application of the scientific method that can enrich the debate. However, if the referee considers that this message is unclear, we are open to modifying the sentence in a way that makes it more transparent.

**"For cases where we obtained the code of a given model, we were not provided with a reason for the license behind it. In fact, in some cases despite getting the code we did not see a license explaining clearly the terms of use." Indeed it would have been crucial to obtain, for every model and every component, the license terms used. That would have helped a lot to discuss accessibility.**

We agree with the referee. Indeed it is the case that probably some models do not have a license. Maybe developers think that in this way no restrictions apply to its distribution, but it is not the case. In some legal systems, releasing software without a license means that it is under the umbrella of a kind of copyright by default. Probably, education on the topic is the only way of overcoming this. Hopefully, the work that we present here helps to make research colleagues more aware of the relevance of the issue and to improve the situation on licensing and code sharing in the future.

---

## Author Response (AR2)

**Reply to editor and reviewers**

Again, we thank the comments by the reviewers. We have prepared a new version of our paper addressing the minor issues pointed out. We have made an effort to tone down the language in this new version. We hope that the new version is suitable to be published in Geosc. Mod. Dev.

**Reply to 'Topical Editor Decision: Publish subject to minor revisions (review by editor) (12 Dec 2020) by Richard Neale'**

**Comments to the Author:**

**Following the second round of reviews this article can be published as is subject to addressing the reviewers' minor revisions. I would certainly ask that you heed them closely as there is a significant amount of speculative and absolutist language used in the paper that still remains from your original submission. This topic is indeed important, representing the core principles that underlie GMD mission that is rare among journals, and so important to GMD. But be aware that GMD reviewers are not social scientists and cannot speak to whether the techniques you have employed in this study are well designed or scientifically accepted. Therefore we are relying on the importance of the subject rather than any absolute certainty in the appropriateness of the methods used.**

**Regards**
**Rich Neale**

Dear Editor,

As requested, we have addressed all the concerns of the reviewers. We have also fixed some minor issues in the manuscript that we have detected during this new iteration. We have removed the term 'climate change models' from the title, that now reads 'climate models'. Moreover, we have attempted to tone down the language and to make less questionable statements.

Many thanks.

**Reply to "Anonymous Referee #1"**

> **Review of "Current status on the need for improved accessibility to climate change models code" by Juan A. Añel and colleagues.**
>
> **This is my second review of the manuscript by Añel et al. As already mentioned in my original review I find the topic of the manuscript to be highly relevant for the scientific community in general and for the climate model community in particular. Personally I still think some of the rather strong statements about the need to make code publicly available for everyone under any circumstances could be framed differently and be better discussed as suggested in my original review but this is up to the authors. Apart from that I found several statements which still need to be clarified as detailed below.**
>
> **Specific Comments:**
>
> **Title: I maintain that CMIP5 models are not "specifically intended" to investigate climate change but also many other aspects of the climate system such as feedbacks, uncertainties, etc even in the absence of climate change (e.g., using the piControl runs). But if the authors are convinced that climate change models is more accurate in the title this is fine.**

We have reflected on this issue again. Indeed, the CMIP5 is probably better known for its contribution to studies on climate change. However, it is also true that Geosc. Mod. Dev. is a journal with a readership that probably knows this difference.
To summarize, we have decided to accept the reviewer's suggestion and remove the word 'change' from the paper title. Probably it does not change the final impact of this work, and the message that we want to transmit will reach the same audience.

> **Page 1, Line 4 & p2, l20: "Climate Model Intercomparison Project (CMIP5)" CMIP5 actually stands for "Coupled Model Intercomparison Project" https://esgf-node.llnl.gov/projects/cmip5/**

Fixed. Thank you for pointing it out.

> **p1, l13-15: For readers (and reviewers) who are not experts in the field of code reproducibility it would be helpful to have a short introduction what CSR means here (and in general) and what makes it so complex. I assume it is not enough to just put code in some repository? And also where does it end? E.g., climate model output is (to my understanding) not bit-by-bit reproducible, what does that mean in this context?**

As suggested, we have included in the text a definition. In this case, we have chosen the one published in 2019 by the U.S. National Academies of Science, Engineering, and Medicine. Also, we cite the work. Now the text reads:

"*CSR, as defined by the U.S. National Academies of Science, Engineering, and Medicine, means 'obtaining consistent results using the same input data, computational methods, and conditions of analysis'*"

We note that in our previous version of the manuscript, although later in the manuscript, we already cited the web page of the ACM where it is defined (ACM. 2018).

**P2, l23-26: I've raised some of these points in my first round of comments already and I'm raising them again here because I think it is crucial to be precise with such rather controversial statements. Without having done any study myself my intuition is to agree with the authors that in several (many) instances climate model code is probably not following an "ideal level of programming practice".**

**BUT: This must not be generalized as the authors do it here. I'm equally convinced that there are climate models out there that can serve at best practice! Therefore I do not think the word "generally", should be used here.**

We have removed the word, as requested by the reviewer.

**Several things the authors should be precise about:**

**- not all climate models are in CMIP (think of a simple 1D energy balance model for educational purposes – this might be perfectly coded, documented and licensed) I assume the authors do not refer to such models here even though they use the generic term "climate model"?**

We consider that this is obvious. Also, in our view, adding a mention to 1D energy balance models, single column, etc. would mess up the readers. Moreover, along the manuscript, we clarify several times that we refer specifically to CMIP5 models. Only those not familiar with the field of climate simulation would doubt about the complexity of such models. For such people, the barrier to understanding our work hardly would be the difference between 1D and GCMs.

Therefore, in this case, we have opted for not including additional explanations in our paper. We hope that our reasoning for it is understood.

**- "ideal level of programming practice" is fairly abstract. The authors introduced the CSR earlier, why not continue to use it (here an in other instances)? For example, if a model is not published how would the authors know if it follows coding and documentation standards? Or is code publication part of the programming practice so that any unpublished model automatically does not follow such practice?**

We do not use CSR because with 'programming practice' we refer to a different thing: Standards on coding, comments and documentation. The programming practice is just one of the multiple barriers or faces involved in full CSR. Failing in programming practice makes it more difficult to comply with CSR, but it could be less compromising than, for example, legal restrictions imposed by a license or do not sharing the code.
However, it is interesting that after several versions of our manuscript, this point can be misunderstood. To make it clear, we have added now a brief explanation in this line, that reads:

"*(i.e., coding standards, number of comments, documentation, etc.)*"

**- "García-Rodríguez et al. (2020) show how programmers have tended to perform very poorly in this regard in particular, and the incidence of comments throughout the code of CMIP5 models is very low." I'm sorry but I'm not able to find any paper under this citation merely a software tool. Am I missing something or is this what the authors want me to look at? In any case I argue it is impossible to state that "the incidence of comments throughout the code of CMIP5 models is very low" as the code is not available for all CMIP5 models (based on the results of this very study!).**

Maybe because this comment by the referee was submitted several months ago, this could be true.

The manuscript pointed out was submitted to SoftwareX for more than one year, and very recently the editors decided that it was out of the scope of the journal. We have just submitted it to IEEE Access. However, the preprint and the Additional Information that contains the evaluation of the code of CMIP5 models are available there. We include here the direct link, in case that the reviewer continues to have problems to access it:
http://fortrananalyser.ephyslab.uvigo.es/docs/AdditionalInformation.pdf

About the second issue, it is true that we have not got access to all the models and that therefore, our statement is a generalization that we can not do. We have rewritten the sentence to be more precise, now it reads:
"*the incidence of comments throughout the code of some CMIP5 models is very low*"

**P3, l9: Maybe the authors can already here link table 1 as it also lists all models involved?**

Done.

**P4, l3-6: I believe the contact information field is available in all NetCDF files for all models as it was required by CMIP5 not only in five models. I would therefore argue that this might even be the preferred place to look at.**
**This is a minor point but if the reason for publishing code is scientific reproducibility, it seems not unreasonable to me to require knowledge of the NetCDF standard from someone who is trying to run a climate model (which arguably requires way more expertise than opening a binary file).**
**Do I understand correctly that the five models mentioned were indeed NOT contacted in the end?**

Again, it is interesting to see how some misunderstandings persist. The reviewer's argument on running the model would be right to study the results' 'replicability'. To assure reproducibility, it is unnecessary to run the model. However, it is necessary to have the tools, and the availability and information to do it. In this way, anyone without need or capability to run a model can verify its computational scientific reproducibility. A completely different thing is assessing the replicability of the results. However, the point is that the models should be available in open repositories without contacting anyone.

About the five models: We have clarified that the contact information is available in the NetCDF files for "another" five models. In this way, we make clear that we look for the information for all the models, but we only find it for five.

About contacting the groups developing these five models: The interpretation by the reviewer is right. We did not reach the groups. In our view, to make additional requests during the review process, with different timelines, would remove part of the value of our analysis, trying to reach all the groups or research centres at the same time and under the same conditions. In our view, they initially failed to communicate on their contact and therefore to make available the code.

**P4, l14: "This analysis is relevant." Delete this sentence?**

Done.

**P3, l25-, Table 1, and Appendix A**
**I've mentioned this in my last review already and the references to the different mails are still unclear to me. Here is an attempted summary:**

**- Mail 1 & 2: "anonymous" requests; given in A1**
**- Mail 3: request explaining the research; this seems to be missing as A2 which is labelled as "Third mail" seems to be something else**
**- Mail 4: this is the survey send if access was denied (but presumably not if there was no answer?); I assume this is the "Third email" in A2? Even though the text in A2 is oddly specific and seems to be taken from a longer exchange??**

To avoid further confusion, now we identify in the text and the caption of Table 1, what Appendix corresponds to each email.

About the text in the third email: The reviewer was right. In our previous version, we included an email very similar to the third one that we sent out after not getting a reply. It was part of the exchange with one of the groups that denied access to the code. As said, the cause of our confusion when including it in the manuscript was the similarity of both emails. We include now the correct version of the email. Thank you for pointing it out.

**In addition table 1 still lists only Mail 1 & 2, it is unclear to which mails that refers.**

As said before, in this version, we clarify it in the caption.

**Reply to Anonymous Referee #2**

We thank to the referee for the insight provided and the relevant questions pointed out.

**I thank the authors for their detailed replies to my comments and acknowledge the rewriting of the conclusions. I still find an historical context about code sharing policy in the climate modelling community would have been valuable, but I understand authors points that it could be the scope of another study (and a lot of work).**
**I suggest to accept the manuscript for publication after some revisions.**

We want to thank the reviewer for their supportive comments. Given that we find the interest in the models' historical development, we have begun to extend our study to earlier CMIP versions. However, at the moment, it is far from a point where we can present sensible results. We hope to publish it in the future.

**The two points that still puzzle me as a reviewer are :**
**1/ Authors refer, in their replies, and also twice in the ms, to Garcia-Rodriguez 2020 manuscript submitted to SoftwareX, which we cannot access to, annihilating my ability to understand/evaluate their point both in their replies and in the text.**

We can not understand what happens with this. In the list of references, we include the webpage to download Garcia-Rodriguez et al. 2020. It is there, it is online, hosted in our university serves. To the best of our knowledge, it has not suffered sensible downtimes. Both the manuscript and the Additional Information (that describe the models' evaluation) are there, linked at the top of the webpage. We include here the direct links to the content, in case it is necessary:
http://fortrananalyser.ephyslab.uvigo.es/docs/softwarex_article_template.pdf
http://fortrananalyser.ephyslab.uvigo.es/docs/AdditionalInformation.pdf

Also, as we point out because of comments by Reviewer#1, the manuscript is now to be submitted to IEEE Access after being rejected in SoftwareX. In the end, after a second review, the editors considered that was not totally in the scope of the journal.

**2/ As in my first review, I am still puzzled with the strength/tone of some statements like "such practice is not acceptable" "cannot be an excuse" that are uncommon in the scientific papers I am used to read/write/review. I guess this particular manuscript aims at making a strong point to improve code sharing in the climate community, and somehow there is a necessity to make such statements. But I feel uncomfortable with such sentences, while confessing being unable to really evaluate their relevance.**

We understand the point by the reviewer. However, sometimes things are black or white. Or we are complying with the scientific or we are not. But there is no such thing as 'we do our best'. Moreover, this depends on human decisions to make information available, not on technical or scientific problems.

That said, we have modified some of the sentences. We think that we can translate the same message without to sound aggressive. Instead of *"such practice is not acceptable"*, now the text reads *"such practice severely undermines the validity of the results"*.

The part of *"cannot be an excuse"* has been rewritten and now reads: *"Barriers to code-sharing through licensing, imposed by, e.g., government bodies, are artificial barriers that depend only on human decisions. They are not due to technical difficulties or scientific reasons."*

**Other minor comment :**

**Page 4, line 23. "This analysis is relevant" I suggest deleting this sentence which seems to be cut from the previous version of the ms.**

Done.

**PAge 7, line 20: "It must be had". Please rephrase.**

Now the text reads 'It must be considered'.

---

## Author Response (AR3)

**UniversidadeVigo**

December 29, 2020

EPhysLab, Universidade de Vigo
Edificio Campus da Auga
32004, Ourense, Spain
Phone: 988 387 235
Email: j.anhel@uvigo.es

Dear Dr. Neale,

Thanks for your comments. This new version of the manuscript includes the corrections requested. Also, we have fixed one misspell and two grammar issues (in Table 2) that we detected after rechecking the text.

Again thank you very much for your consideration.

On behalf of all the coauthors,

Dr. Juan A. Añel
Associate Professor
EPhysLab, Universidade de Vigo
http://ephyslab.uvigo.es/juan/en/